# Autodecoding Latent 3D Diffusion Models

**Evangelos Ntavelis**\*
Computer Vision Lab
ETH Zurich
Zürich, Switzerland
entavelis@vision.ee.ethz.ch

**Aliaksandr Siarohin**
Creative Vision
Snap Inc.
Santa Monica, CA, USA
asiarohin@snapchat.com

**Kyle Olszewski**
Creative Vision
Snap Inc.
Santa Monica, CA, USA
kolszewski@snap.com

**Chaoyang Wang**
CI2CV Lab
Carnegie Mellon University
Pittsburgh, PA, USA
chaoyanw@cs.cmu.edu

**Luc Van Gool**
CVL, ETH Zurich, CH
PSI, KU Leuven, BE
INSAIT, Un. Sofia, BU
vangool@vision.ee.ethz.ch

**Sergey Tulyakov**
Creative Vision
Snap Inc.
Santa Monica, CA, USA
stulyakov@snapchat.com

## Abstract

We present a novel approach to the generation of static and articulated 3D assets that has a 3D *autodecoder* at its core. The 3D *autodecoder* framework embeds properties learned from the target dataset in the latent space, which can then be decoded into a volumetric representation for rendering view-consistent appearance and geometry. We then identify the appropriate intermediate volumetric latent space, and introduce robust normalization and de-normalization operations to learn a 3D diffusion from 2D images or monocular videos of rigid or articulated objects. Our approach is flexible enough to use either existing camera supervision or no camera information at all – instead efficiently learning it during training. Our evaluations demonstrate that our generation results outperform state-of-the-art alternatives on various benchmark datasets and metrics, including multi-view image datasets of synthetic objects, real in-the-wild videos of moving people, and a large-scale, real video dataset of static objects.

Code & Visualizations: https://github.com/snap-research/3DVADER

## 1 Introduction

Photorealistic generation is undergoing a period that future scholars may well compare to the enlightenment era. The improvements in quality, composition, stylization, resolution, scale, and manipulation capabilities of images were unimaginable just over a year ago. The abundance of online images, often enriched with text, labels, tags, and sometimes per-pixel segmentation, has significantly accelerated such progress. The emergence and development of denoising diffusion probabilistic models (DDPMs) [75, 77, 28] propelled these advances in image synthesis [53, 78, 80, 76, 17, 18, 63, 36, 86] and other domains, *e.g.* audio ([10, 20, 95]) and video ([24, 89, 82, 29, 25, 47]).

However, the world is 3D, consisting of static and dynamic objects. Its geometric and temporal nature poses a major challenge for generative methods. First of all, the data we have consists mainly of images and monocular videos. For some limited categories of objects, we have 3D meshes with corresponding multi-view images or videos, often obtained using a tedious capturing process or created manually by artists. Second, unlike CNNs, there is no widely accepted 3D or 4D representation suitable for 3D geometry and appearance generation. As a result, with only a few exceptions [74], most of the existing 3D generative methods are restricted to a narrow range of object

---

\*Work done during internship at Creative Vision Team - Snap Inc

37th Conference on Neural Information Processing Systems (NeurIPS 2023).

categories, suitable to the available data and common geometric representations. Moving, articulated objects, *e.g.* humans, compound the problem, as the representation must also support deformations.

In this paper, we present a novel approach to designing and training denoising diffusion models for 3D-aware content suitable for efficient usage with datasets of various scales. It is generic enough to handle both rigid and articulated objects. It is versatile enough to learn diverse 3D geometry and appearance from multi-view images and monocular videos of both static and dynamic objects. Recognizing the poses of objects in such data has proven to be crucial to learning useful 3D representations [6, 7, 73, 74]. Our approach is thus designed to be robust to the use of ground-truth poses, those estimated using structure-from-motion, or using no input pose information at all, but rather learning it effectively during training. It is scalable enough to train on single- or multi-category datasets of large numbers of diverse objects suitable for synthesizing a wide range of realistic content.

Recent diffusion methods consist of two stages [63]. During the first stage, an autoencoder learns a rich latent space. To generate new samples, a diffusion process is trained during the second stage to explore this latent space. To train an image-to-image autoencoder, many images are needed. Similarly, training 3D autoencoders requires large quantities of 3D data, which is very scarce. Previous works used synthetic datasets such as ShapeNet [8] (DiffRF[49], SDFusion [12], *etc.*), and were thus restricted to domains where such data is available.

In contrast to these works, we propose to use a volumetric auto*decoder* to learn the latent space for diffusion sampling. In contrast to the autoencoder-based approach, our autodecoder maps a 1D vector to each object in the training set, and thus does not require 3D supervision. The autodecoder learns 3D representations from 2D observations, using rendering consistency as supervision. Following UVA [70] this 3D representation supports the articulated parts necessary to model non-rigid objects.

There are several key challenges with learning such a rich, latent 3D space with an autodecoder. First, our autodecoders do not have a clear "bottleneck." Starting with a 1D embedding, they upsample it to latent features at many resolutions, until finally reaching the output radiance and density volumes. Here, each intermediate volumetric representation could potentially serve as a "bottleneck." Second, autoencoder-based methods typically regularize the bottleneck by imposing a KL-Divergence constraint [38, 63], meaning diffusion must be performed in this regularized space.

To identify the best intermediate representation to perform diffusion, one can perform exhaustive layer-by-layer search. This, however, is very computationally expensive, as it requires running hundreds of computationally expensive experiments. Instead, we propose robust normalization and denormalization operations which can be applied to any layers of a pre-trained and fixed autodecoder. These operations compute robust statistics to perform layer normalization and, thus, allow us to train the diffusion process at any intermediate resolution of the autodecoder. We find that at fairly low resolutions, the space is compact and provides the necessary regularization for geometry, allowing the training data to contain only sparse observations of each object. The deeper layers, on the other hand, operate more as upsamplers. We provide extensive analysis to find the appropriate resolution for our autodecoder-based diffusion techniques.

We demonstrate the versatility and scalability of our approach on various tasks involving rigid and articulated 3D object synthesis. We first train our model using multi-view images and cameras in a setting similar to DiffRF [49] to generate shapes of a limited number of object categories. We then scale our model to hundreds of thousands of diverse objects train using the real-world MVImgNet [92] dataset, which is beyond the capacity of prior 3D diffusion methods. Finally, we train our model on a subset of CelebV-Text [90], consisting of ∼44K sequences of high-quality videos of human motion.

## 2 Related Work

### 2.1 Neural Rendering for 3D Generation

Neural radiance fields, or NeRFs (Mildenhall et al., 2020 [48]), enable high-quality novel view synthesis (NVS) of rigid scenes learned from 2D images. Its approach to volumetric neural rendering has been successfully applied to various tasks, including *generating* objects suitable for 3D-aware NVS. Inspired by the rapid development of generative adversarial models (GANs) [22] for generating 2D images [22, 5, 32, 33, 35, 34] and videos [79, 72, 91], subsequent work extends them to 3D content generation with neural rendering techniques. Such works [67, 51, 54, 52, 88] show promising

results for this task, yet suffer from limited multi-view consistency from arbitrary viewpoints, and experiencing difficulty in generalizing to multi-category image datasets.

A notable work in this area is pi-GAN (Chan et al., 2021 [6]), which employs neural rendering with periodic activation functions for generation with view-consistent rendering. However, it requires a precise estimate of the dataset camera pose distribution, limiting its suitability for free-viewpoint videos. In subsequent works, EG3D (Chan et al., 2022 [7]) and EpiGRAF (Skorokhodov et al. [73]) use tri-plane representations of 3D scenes created by a generator-discriminator framework based on StyleGAN2 (Karras et al., 2020 [35]). However, these works require pose estimation from keypoints (*e.g.* facial features) for training, again limiting the viewpoint range.

These works primarily generate content within one object category with limited variation in shape and appearance. A notable exception is 3DGP [74], which generalizes to ImageNet [15]. However, its reliance on monocular depth prediction limits it to generating front-facing scenes. These limitations also prevent these approaches from addressing deformable, articulated objects. In contrast, our method is applicable to both deformable and rigid objects, and covers a wider range of viewpoints.

## 2.2 Denoising Diffusion Modeling

Denoising diffusion probabilistic models (DDPMs) [75, 28] represent the generation process as the learned denoising of data progressively corrupted by a sequence of diffusion steps. Subsequent works improving the training objectives, architecture, and sampling process [28, 17, 86, 36, 63, 53, 76] demonstrated rapid advances in high-quality data generation on various data domains. However, such works have primarily shown results for tasks in which samples from the target domain are fully observable, rather than operating in those with only partial observations of the dataset content.

One of the most important of such domains is 3D data, which is primarily observed in 2D images for most real-world content. Some recent works have shown promising initial results in this area. DiffRF [49] proposes reconstructing per-object NeRF volumes for synthetic datasets, then applying diffusion training on them within a U-Net framework. However, it requires the reconstruction of many object volumes, and is limited to low-resolution volumes due to the diffusion training's high computational cost. As our framework instead operates in the latent space of the autodecoder, it effectively shares the learned knowledge from all training data, thus enabling low-resolution, latent 3D diffusion. In [12], a 3D autoencoder is used for generating 3D shapes, but this method require ground-truth 3D supervision, and only focuses on shape generation, with textures added using an off-the-shelf method [60]. In contrast, our framework learns to generate the surface appearance and corresponding geometry without such ground-truth 3D supervision.

Many recent works [2, 68, 23, 9] combine a denoising diffusion approach with a tri-plane representation [7] for 3D generation. They perform diffusion on the embedding vector of an autodecoder [2], the bottleneck of an autoencoder [23], or directly, on a pre-computed [68] or a simultaneously learned tri-plane [9]. Nevertheless, these works focus on small datasets or require a dense point clouds and ground truth object meshes, which are not readily available for real-object image datasets. The triplane representation requires an MLP decoder that substantially increases the volumetric rendering time. Our voxel-decoder does not has such a requirement as it directly outputs color and density, and thus permitting faster training on large-scale real image datasets.

Recently, several works [60, 43, 11] propose using large-scale, pre-trained text-to-image 2D diffusion models for 3D generation. The key idea behind these methods is to use 2D diffusion models to evaluate the quality of renderings from randomly sampled viewpoints, then use this information to optimize a 3D-aware representation of the content. Compared to our method, however, such approaches require a far more expensive optimization process to generate each novel object.

## 3 Methodology

Our method is a two-stage approach. In the first stage, we learn an autodecoder $G$ containing a library of embedding vectors corresponding to the objects in the training dataset. These vectors are first processed to create a low-resolution, latent 3D feature volume, which is then progressively upsampled and finally decoded into a voxelized representation of the generated object's shape and appearance. This network is trained using volumetric rendering techniques on this volume, with 2D reconstruction supervision from the training images.

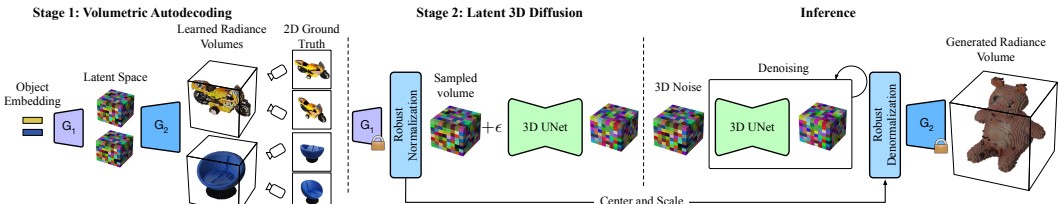

Figure 1: **Our proposed two-stage framework**. Stage 1 trains an autodecoder with two generative components, $G_1$ and $G_2$. It learns to assign each training set object a 1D embedding that is processed by $G_1$ into a latent volumetric space. $G_2$ decodes these volumes into larger radiance volumes suitable for rendering. Note that we are using only 2D supervision to train the autodecoder. In Stage 2, the autodecoder parameters are frozen. Latent volumes generated by $G_1$ are then used to train the 3D denoising diffusion process. At inference time, $G_1$ is not used, as the generated volume is randomly sampled, denoised, and then decoded by $G_2$ for rendering.

During the second stage, we split the autodecoder $G$ into two parts, $G = G_2 \circ G_1$. We then employ this autodecoder to train a 3D diffusion model operating in the compact, 3D latent space obtained from $G_1$. [2] Using the structure and appearance properties extracted from the autodecoder training dataset, this 3D diffusion process allows us to use this network to efficiently generate diverse and realistic 3D content. The full pipeline is depicted in Fig. 1.

Below, we first describe the volumetric autodecoding architecture (Sec. 3.1). We then describe the training procedure and reconstruction losses for the autodecoder (Sec. 3.2). Finally, we provide details for our training and sampling strategies for 3D diffusion in the decoder's latent space (Sec. 3.3).

## 3.1 Autodecoder architecture

**Canonical Representation.** We use a 3D voxel grid to represent the 3D structure and appearance of an object. We assume the objects are in their canonical pose, such that the 3D representation is decoupled from the camera poses. This decoupling is necessary for learning compact representations of objects, and also serves as a necessary constraint to learn meaningful 3D structure from 2D images without direct 3D supervision. Specifically, the canonical voxel representation consists of a density grid $V^{\text{Density}} \in \mathbb{R}^{S^3}$ which is a discrete representation of the density field with resolution $S^3$, and $V^{RGB} \in \mathbb{R}^{S^3 \times 3}$ which represents the RGB radiance field. We employ volumetric rendering, integrating the radiance and opacity values along each view ray similar to NeRFs [48]. In contrast to the original NeRF, however, rather than computing these local values using an MLP, we tri-linearly interpolate the density and RGB values from the decoded voxel grids, similar to Plenoxels [64].

**Voxel Decoder.** The 3D voxel grids for density and radiance, $V^{\text{Density}}$ and $V^{\text{RGB}}$, are generated by a volumetric autodecoder $G$ that is trained using rendering supervision from 2D images. We choose to directly generate $V^{\text{Density}}$ and $V^{\text{RGB}}$, rather than intermediate representations such as feature volumes or tri-planes, as it is more efficient to render and ensures consistency across multiple views. Note that feature volumes and tri-planes require running an MLP pass for each sampled point, which requires significant computational cost and memory during training and inference.

The decoder is learned in the manner of GLO [4] across various object categories from large-scale multi-view or monocular video datasets. The architecture of our autodecoder is adapted from [70]. However, in our framework we want to support large scale datasets which poses a challenge in designing the decoder architecture with the capability to generate high-quality 3D content across various categories. In order to represent each of the $\sim$300K objects in our largest dataset we need very high-capacity decoder. As we found the relatively basic decoder of [70] produced poor reconstruction quality, we introduce the following key extensions:

- To support the diverse shapes and appearances in our target datasets, we find it crucial to increase the length of the embedding vectors learned by our decoder from 64 to 1024.

---

[2]We experimented with diffusion at different feature volume resolutions, ranging from $4^3$ at the earliest stage to $16^3$ in the later stages. These results are described in our evaluations (Sec. 4.3, Fig. 5).

- We increase the number of residual blocks at each resolution in the autodecoder from 1 to 4.
- Finally, to harmonize the appearance of the reconstructed objects we introduce self-attention layers [81] in the second and third levels (resolutions $8^3$ and $16^3$).

**Scaling the Embedding Codebook for Large Datasets.** Each object in the training set is encoded by an embedding vector. However, storing a separate vector for each object is burdensome, especially for large datasets. As such, we propose a technique to significantly reduce the parameter footprint of our embeddings, while allowing effective generation from large-scale datasets.

Similar to StyleGenes' approach [55], we combine smaller embedding *subvectors* to create unique per-object vectors. The decoder's input is a per-object embedding vector $H_k \in \mathbb{R}^l$ with length $l_v$. It is a concatenation of smaller subvectors $h_i^j$, where each subvector is selected from an ordered codebook with $n_c$ entries, with each entry containing collection of $n_h$ embedding vectors of length $l_v/n_c$:

$$H_k = \left[ h_1^{k_1}, h_2^{k_2}, ..., h_{n_c}^{k_{n_c}} \right], \tag{1}$$

where $k_i \in \{1, 2, ..., n_h\}$ is the set of indices used to select from the $n_h$ possible codebook entries for position $i$ in the final vector. This method allows for exponentially more combinations of embedding vectors, greatly reducing the number of stored parameters compared to a single embedding codebook. In contrast to [55], the index $j$ for the vector $h_i^j$ at position $i$ is not randomly selected for each position to access its corresponding codebook entry. We use a *hashing function* [16] to map each training object index $k$ to its corresponding embedding index.

## 3.2 Autodecoder Training

We train the decoder from image data through analysis-by-synthesis, with the primary objective of minimizing the difference between the decoder's rendered images and the training images. We render RGB color image $C$ using volumetric rendering [48], additionally in order to supervise silhouette of the objects we render 2D occupancy mask $O$.

**Pyramidal Perceptual Loss.** As in [69, 70], we employ a pyramidal perceptual loss based on [31] on the rendered images as our primary reconstruction loss:

$$\mathcal{L}_{\text{rec}}(\hat{C}, C) = \sum_{l=0}^{L} \sum_{i=0}^{I} \left| \text{VGG}_i(\text{D}_l(\hat{C})) - \text{VGG}_i(\text{D}_l(C)) \right|, \tag{2}$$

where $\hat{C}, C \in [0, 1]^{H \times W \times 3}$ are the RGB rendered and training images of resolution $H \times W$, respectively; $\text{VGG}_i$ is the $i^{\text{th}}$-layer of a pre-trained VGG-19 [71] network; and operator $D_l$ downsamples images to the resolution for pyramid level $l$.

**Foreground Supervision.** Since we only interested in modeling single objects, in all the datasets considered in this work, we remove the background. However if the color of the object is black (which corresponds to the absence of density), the network can make the object semi-transparent. To improve the overall shape of the reconstructed objects, we make use of a foreground supervision loss. Using binary foreground masks (estimated by an off-the-shelf matting method [44], Segment Anything [39] or synthetic ground-truth masks, depending on the dataset), we apply an L1 loss on the rendered occupancy map to match that of the mask corresponding to the image.

$$\mathcal{L}_{\text{seg}}(\hat{O}, O) = \frac{1}{HW} \|O - \hat{O}\|_1, \tag{3}$$

where $\hat{O}, O \in [0, 1]^{H \times W}$ are the inferred and ground-truth occupancy masks, respectively. We provide visual comparison of the inferred geometry for this loss in the supplement.

**Multi-Frame Training.** Because our new decoder have a large capacity, generating a volume incur much larger overhead compared to rendering an image based on this volume (which mostly consists of tri-linear sampling of the voxel cube). Thus, rather than rendering a single view for the canonical representation of the target object in each batch, we instead render 4 views for each object in the batch. This technique incurs no significant overhead, and effectively increases the batch size four times. As

an added benefit, we find that this technique improves on the overall quality of the generated results, since it significantly reduce batch variance. We ablate this technique and our key architectural design choices, showing their effect on the sample quality (Sec. 4.3, Tab. 2).

**Learning Non-Rigid Objects.** For articulated, non-rigid objects, *e.g.* videos of human subjects, we must model a subject's shape and local motion from dynamic poses, as well as the corresponding non-rigid deformation of local regions. Following [70], we assume these sequences can be decomposed into a set of $N_p$ smaller, rigid components (10 in our experiments) whose poses can be estimated for consistent alignment in the canonical 3D space. The camera poses for each component are estimated and progressively refined during training, using a combination of learned 3D keypoints for each component of the depicted subject and the corresponding 2D projections predicted in each image. This estimation is performed via a differentiable Perspective-n-Point (PnP) algorithm [40].

To combine these components with plausible deformations, we employ a learned volumetric linear blend skinning (LBS) operation. We introduce a voxel grid $V^{LBS} \in \mathbb{R}^{S^3 \times N_p}$ to represent the skinning weights for each deformation components. As we assume no prior knowledge about the content or assignment of object components, the skinning weights for each component are also estimated during training. Please see the supplement for additional details.

## 3.3 Latent 3D Diffusion

**Architecture.** Our diffusion model architecture extends prior work on diffusion in a 2D space [36] to the latent 3D space. We implement its 2D operations, including convolutions and self-attention layers, in our 3D decoder space. In the text-conditioning experiments, after the self-attention layer, we use a cross-attention layer similar to that of [63]. Please see the supplement for more details.

**Feature Processing.** One of our key observation is that the features $F$ in the latent space of the 3D autodecoder have a bell-shaped distribution (see the supplement), which eliminates the need to enforce any form of prior on it, *e.g.* as in [63]. Operating in the latent space without a prior enables training a single autodecoder for each of the possible latent diffusion resolutions. However, we observe that the feature distribution $F$ has very long tails. We hypothesise this is because the final density values inferred by the network do not have any natural bounds, and thus can fall within any range. In fact, the network is encouraged to make such predictions, as they have the sharpest boundaries between the surface and empty regions. However, to allow for a uniform set of diffusion hyper-parameters for all datasets and all trained autodecoders, we must normalize their features into the same range. This is equivalent to computing the center and the scale of the distribution. Note that, due to the very long-tailed feature distribution, typical mean and standard deviation statistics will be heavily biased. We thus propose a robust alternative based on the feature distribution quantiles. We take the *median* $m$ as the center of the distribution and approximate its scale using the Normalized InterQuartile Range (IQR) [85] for a normal distribution: $0.7413 \times IQR$. Before using the features $F$ for diffusion, we normalize them to $\hat{F} = \frac{(F-m)}{IQR}$. During inference, when producing the final volumes we de-normalize them as $\hat{F} \times IQR + m$. We call this method *robust normalization*. Please see the supplement for an evaluation of its impact.

**Sampling for Object Generation.** During inference we rely on the sampling method from EDM [36], with several slight modifications. We fix EDM's hyperparameter matching the dataset's distribution to 0.5 regardless of the experiment, and modify the feature statistics in our feature processing step. We also introduce classifier free guidance [27] for our text-conditioning experiments (Sec. 4.5). We found that setting the weight equal to 3 yields good results across all datasets.

## 4 Results and Evaluations

In this section, we evaluate our method on multiple diverse datasets (see Sec. 4.1) for both unconditional 4.2 and conditional settings 4.5. We also ablate the design choices in our autodecoder and diffusion in Secs. 4.3 and 4.4, respectively.

| Method | PhotoShape Chairs [57] | | ABO Tables [13] | |
|---|---|---|---|---|
| | FID ↓ | KID ↓ | FID ↓ | KID ↓ |
| $\pi$-GAN [6] | 52.71 | 13.64 | 41.67 | 13.81 |
| EG3D [7] | 16.54 | 8.412 | 31.18 | 11.67 |
| DiffRF [49] | 15.95 | 7.935 | 27.06 | 10.03 |
| Ours | **11.28** | **4.714** | **18.44** | **6.854** |

Table 1: Results on the synthetic PhotoShape Chairs [57] and ABO Tables [13] datasets. Overall, our method outperforms state-of-the-art GAN-based and diffusion-based approaches. KID scores are multiplied by $10^3$.

| Model Variant | PSNR ↑ | LPIPS ↓ |
|---|---|---|
| Ours | **27.719** | **6.255** |
| - Multi-Frame Training | 27.176 | 6.855 |
| - Self-Attention | 27.335 | 6.738 |
| - Increased Depth | 27.24 | 6.924 |
| - Embedding Length ($1024 \rightarrow 64$) | 25.985 | 8.332 |

Table 2: Our 3D autodecoder ablation results. "-" indicates this component has been removed. As we remove each sequentially, the top row depicts results for the unmodified architecture and training procedure. LPIPS results are multiplied by $10^2$.

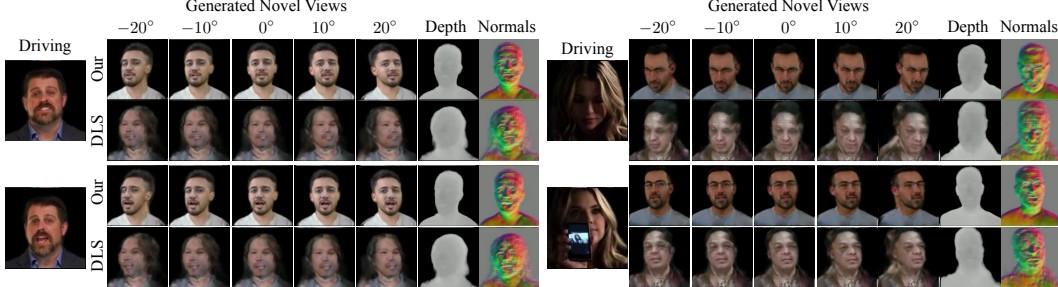

Figure 2: Qualitative comparisons with Direct Latent Sampling (DLS) [70] on CelebV [90]. We show the two driving videos for two random identities: the top identity in each block is generated by our method, the bottom identity in each block is generated by DLS [70]. We also show the rendered depth and normals.

## 4.1 Datasets and Data Processing

Below we describe the datasets used for our evaluations. We mostly evaluate our method on datasets of synthetic renderings of 3d objects [13, 57, 14]. However, we also provide results on a challenging video dataset of dynamic human subjects [90] and dataset of static object videos [92].

**ABO Tables.** Following [49], we evaluate our approach on renderings of objects from the Tables subset of the Amazon Berkeley Objects (ABO) dataset [13], consisting of $1,676$ training sequences with 91 renderings per sequence, for a total of $152,516$ renderings.

**PhotoShape Chairs.** Also as in [49], we use images from the Chairs subset of the PhotoShape dataset [57], totaling $3,115,200$ frames, with 200 renderings for each of $15,576$ chair models.

**Objaverse.** This dataset [14] contains ∼800K publicly available 3D models. As the of the object geometry and appearance varies, we use a manually-filtered subset of ∼300K unique objects (see supplement for details). We render 6 images per training object, for a total of ∼1.8 million frames.

**MVImgNet.** For this dataset [92], we use ∼6.5 million frames from $219,188$ videos of real-world objects from 239 categories, with an average of 30 frames each. We use Grounded Segment Anything [45, 39] for background removal, then apply filtering (see supplement) to remove objects with failed segmentation. This process results in $206,990$ usable objects.

**CelebV-Text.** The CelebV-Text dataset [90] consists of ∼70K sequences of high-quality videos of celebrities captured in in-the-wild environments, lighting, motion, and poses. They generally depict the head, neck, and upper-torso region, but contain more challenging pose and motion variation than prior datasets, *e.g.* VoxCeleb [50]. We use the robust video matting framework of [44] to obtain our masks for foreground supervision (Sec. 3.2). Some sample filtering (described in the supplement) was needed for sufficient video quality and continuity for training. This produced ∼44.4K unique videos, with an average of ∼ 373 frames each, totaling ∼16.6M frames.

For training, we use the **camera parameters** used to render each synthetic object dataset, and the estimated parameters provided for the real video sequences in MVImgNet, adjusted to center and scale the content to our rendering volume, (see supplement for details). For the human videos in CelebV-Text, we train an additional pose estimator along with the autodecoder $G$ to predict poses for each articulated region per frame, such that all objects can be aligned in the canonical space (Sec. 3.2). Note that for creating dynamic 3D video, we can use sequences of poses transferred from the real video of another person from the dataset.

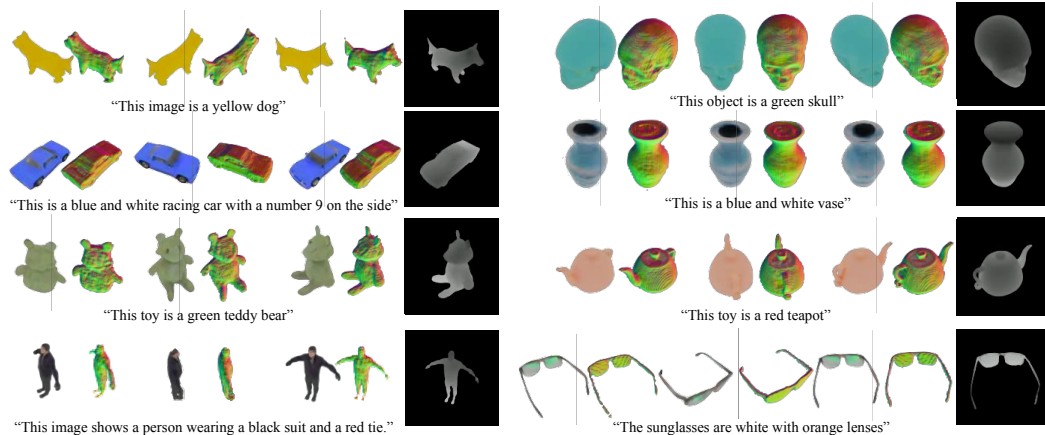

Figure 3: We show generated samples of our model trained using rendered images from Objaverse [14]. We show three views for each object, along with the normals for each view. We also show depth for the right-most view. Text-conditioned results are shown. Grouth-truth captions are generated by MiniGPT-4 [94].

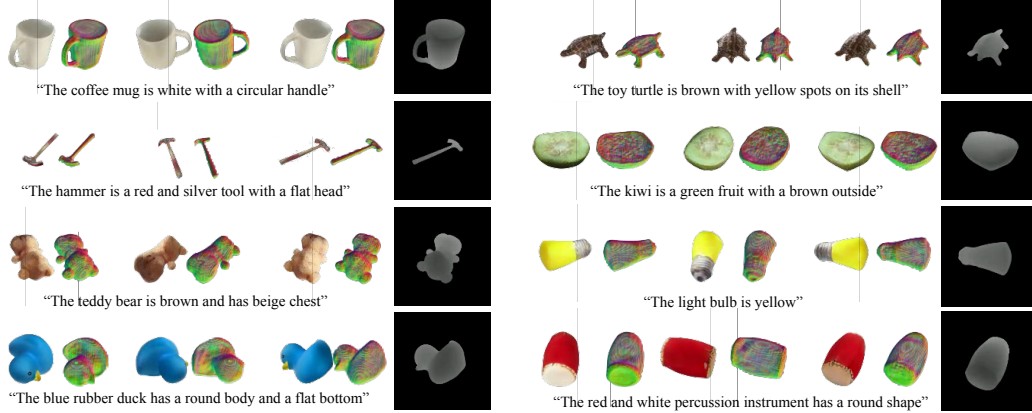

Figure 4: We show generated samples from our model trained using monocular videos from MVImgNet [92]. We show three views for each object, along with the normals for each view. We also show depth for the right-most view. Text-conditioned results are shown. Ground-truth captions are generated by MiniGPT-4 [94].

### 4.2 Unconditional Image Generation

**Synthetic Datasets.** Following the evaluation protocol of [49], we report results on the ABO Tables and PhotoShape Chairs datasets. These results on single-category, synthetically rendered datasets that are relatively small compared to the others, demonstrate that our approach also performs well with smaller, more homogeneous data. We render 10 views of 1K samples from each dataset, and report the Frèchet Inception Distance (FID) [26] and Kernel Inception Distance (KID) [3] when compared to 10 randomly selected ground-truth images from each training sequence. We report the results compared to both GAN-based [6, 7] and more recent diffusion-based approaches [49] methods, as seen in Tab. 1. We see that our method significantly outperforms state-of-the-art methods using both metrics on the Tables dataset, and achieves better or comparable results on the Chairs dataset.

**Large-Scale Datasets.** We run tests on the large-scale datasets described above: MVImgNet, CelebV-Text and Objaverse. For each dataset, we render 5 images from random poses for each of 10K generated samples. We report the FID and KID for these experiments compared to 5 ground-truth images for each of 10K training objects. As no prior work demonstrates the ability to generalize to such large-scale datasets, we compare our model against directly sampling the 1D latent space of our base autodecoder architecture (using noise vectors generated from a standard normal distribution). This method of 3D generation was shown to work reasonably well [70]. We also evaluate our approach with different numbers of diffusion steps (16, 32 and 64). The results can be seen in Tab. 3. Visually, we compare with [70] in Fig. 2. Our qualitative results show substantially higher fidelity, quality of geometry and texture. We can also see that when identities are sampled directly in the 1D

latent space, the normals and depth are significantly less sharp, indicating that there exist spurious density in the sampled volumes. Tab. 3 further supports this observation: both the FID and KID are significantly lower than those from direct sampling, and generally improve with additional steps.

## 4.3 Autodecoder Ablation

We conduct an ablation study on the key design choices for our autodecoder architecture and training. Starting with the final version, we subtract the each component described in Sec. 3.1. We then train a model on the PhotoShape Chairs dataset and render 4 images for each of the ~15.5K object embeddings.

Tab. 2 provides the the PSNR [30] and

|                          | CelebV-Text [90] | | MVImgNet [92] | | Objaverse [14] | |
|--------------------------|---------|---------|---------|---------|---------|---------|
| Method                   | FID↓    | KID↓    | FID↓    | KID↓    | FID↓    | KID↓    |
| Direct Latent Sampling [70] | 69.21 | 73.74 | 97.51 | 69.22 | 72.76 | 53.68 |
| Ours - *16 Steps*        | **48.01** | 49.49 | 62.21 | 39.94 | 47.49 | 32.44 |
| Ours - *32 Steps*        | 49.74 | **46.2** | 51.26 | 28.45 | 43.68 | 31.7 |
| Ours - *64 Steps*        | 50.27 | 47.72 | **43.85** | **23.91** | **40.49** | **29.37** |

Table 3: Results on large-scale multi-view image (Objaverse [14] & MVImgNet [92]) and monocular video (CelebV-Text [90]) datasets. The KID score is multiplied by $10^3$.

LPIPS [93] reconstruction metrics. We find that the final version of our process significantly outperforms the base architecture [70] and training process. While the largest improvement comes from our increase in the embedding size, we see that simply removing the multi-frame training causes a noticeable drop in quality by each metric. Interestingly, removing the self-attention layers marginally increases the PSNR and lowers the LPIPS. This is likely due to the increased complexity in training caused by these layers, which for a dataset of this size, may be unnecessary. For large-scale datasets, we observed significant improvement with this feature. Both decreasing the depth of the residual convolution blocks and reducing the embedding size cause noticeable drops in the overall quality, particularly the latter. This suggests that the additional capacity provided by these components is impactful, even on a smaller dataset.

## 4.4 Diffusion Ablation

We also perform ablation on our diffusion process, evaluating the effect of the choice of the number of diffusion steps (16, 32, and 64), and the autodecoder resolution at which we perform diffusion ($4^3$, $8^3$, and $16^3$). For these variants, we follow the generation quality training and evaluation protocol on the PhotoShape Chairs (Sec. 4.2), except that we disable stochasticity in our sampling during inference for more consistent performance across these tests. Each model was trained using roughly the same amount of time and computation. Fig. 5 shows the results. Interestingly, we can see a clear distinction between the results obtained from

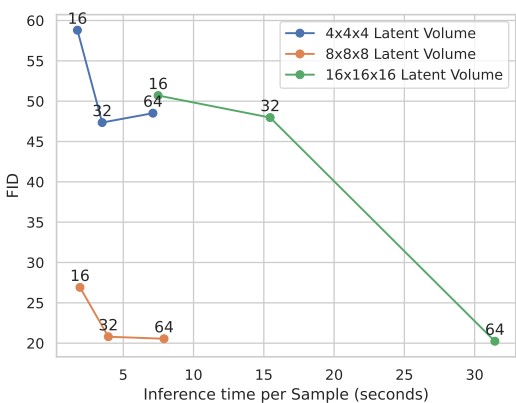

Figure 5: Impact of diffusion resolution and number of sampling steps on sample quality and inference time.

diffusion at the earlier or later autodecoder stages, and those from our the results with resolution $8^3$. We hypothesize that at lowest resolution layers overfit to the training dataset, thus when processing novel objects via diffusion, the quality degrades significantly. Training at a higher resolution requires substantial resources, limiting the convergence seen in a reasonable amount of time. The number of sampling steps has a smaller, more variable impact. Going from 16 to 32 steps improves the results with a reasonable increase in inference time, but at 64 steps, the largest improvement is at the $16^3$ resolution, which requires more than 30 seconds per sample. Our chosen diffusion resolution of $8^3$ achieves the best results, allowing for high sample quality at 64 steps (used in our other experiments) with only ~8 seconds of computation, but provides reasonable results with 32 steps in ~4 seconds.

## 4.5 Conditional Image Generation

Finally, we train diffusion models with text-conditioning. For MVImgNet and Objaverse, we generate the text with an off-the-shelf captioning system [94]. Qualitative results for MVImgNet and Objaverse are in Figs. 4 and 3. We observe that in all cases, our method generates objects with reasonable geometry that generally follow the prompt. However, some details can be missing. We believe our

model learns to ignore certain details from text prompts, as MiniGPT-4 often hallucinates details inconsistent with the object's appearance. Better captioning systems should help alleviate this issue.

### 4.6 Design Choices for Large-Scale 3D Object Synthesis

The goal of our work is to enable 3D object synthesis by training a model on large and diverse multi-view image datasets. To realize this goal there are two main design choices that we need to make: (a) what is the appropriate 3D representation and (b) generative modelling approach.

Recent works [2, 68, 23, 9] use tri-planes as their 3D representation. However, when the multi-view supervision is scarce and ground truth camera information is not available, such as in video datasets like CelebV, tri-planes tend to degrade to prediction of flat objects [70]. Moreover, tri-planes require an additional MLP for volumetric rendering, applied for every of the 128 ray point samples and $128^2$ output pixels we use in our setting. In contrast, our voxel grid autodecoder outputs directly color and density. Tri-planes are faster to *autodecode*, but rendering them is much slower. Training for an iteration with our $64^3$ voxel grid takes $0.22s$. Tri-planes of size $64^2$, $128^2$, $256^2$ and $512^2$ require $0.33, 0.33, 0.38, 0.46$ seconds respectively. We use 32 channels per plane, a two-layer MLP with 32 hidden channels, and a 2D autodecoder. This can severely affect the training time for a large dataset.

EG3D [7] and GET3D [21] propose an adversarial approach to 3D synthesis. Both, base their generators on StyleGAN [33], which for 2D datasets requires considerable changes to produce good results in large and diverse datasets [65]. Training both on Objaverse, we find they fail to converge, as seen in Fig. 11. Thus, we believe our diffusion-based approach is better suited for our goal.

## 5   Conclusion

Despite the inherent challenges in performing flexible 3D content generation for arbitrary content domains without 3D supervision, our work demonstrates this is possible with the right approach. By exploiting the inherent power of autodecoders to synthesize content in a domain without corresponding encoded input, our method learns representations of the structure and appearance of diverse and complex content suitable for generating high-fidelity 3D objects using only 2D supervision. Our latent volumetric representation is conducive to 3D diffusion modeling for both conditional and unconditional generation, while enabling view-consistent rendering of the synthesized objects. As seen in our results, this generalizes well to various types of domains and datasets, from relatively small, single-category, synthetic renderings to large-scale, multi-category real-world datasets. It also supports the challenging task of generating articulated moving objects from videos. No prior work addresses each of these problems in a single framework. The progress shown here suggests there is potential to develop and extend our approach to address other open problems.

**Limitations.**   While we demonstrate impressive and state-of-the-art results on diverse tasks and content, several challenges and limitations remain. Here we focus on images and videos with foregrounds depicting one key person or object. The generation or composition of more complex, multi-object scenes is a challenging task and an interesting direction for future work. As we require multi-view or video sequences of each object in the dataset for training, single-image datasets are not supported. Learning the appearance and geometry of diverse content for controllable 3D generation and animation from such limited data is quite challenging, especially for articulated objects. However, using general knowledge about shape, motion, and appearance extracted from datasets like ours to reduce or remove the multi-image requirement when learning to generate additional object categories may be feasible with further exploration. This would allow the generation of content learned from image datasets of potentially unbounded size and diversity.

**Broader Impact.**   Our work shares similar concerns with other generative modeling efforts, *e.g.*, potential exploitation for misleading content. As with all such learning-based methods, biases in training datasets may be reflected in the generated content. Appropriate caution must be applied when using this method to avoid this when it may be harmful, *e.g.* human generation. Care must be taken to only use this method on public data, as the privacy of training subjects may be compromised if our framework is used to recover their identities. The environmental impact of methods requiring substantial energy for training and inference is also a concern. However, our approach makes our tasks more tractable by removing the need for the curation and processing of large-scale 3D datasets, and is thus more amenable to efficient use than methods requiring such input.

**Acknowledgements** We would like to thank Michael Vasilkovsky for preparing the ObjaVerse renderings, and Colin Eles for his support with infrastructure. Moreover, we would like to thank Norman Müller, author of DiffRF paper, for his invaluable help with setting up the DiffRF baseline, the ABO Tables and PhotoShape Chairs datasets, and the evaluation pipeline as well as answering all related questions. A true marvel of a scientist. Finally, Evan would like to thank Claire and Gio for making the best cappuccinos and fueling up this research.

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

# A  Additional Experiments and Results

## A.1  Geometry Generation Evaluation

Following the point cloud evaluation protocol of [1], we measure the Coverage Score (COV) and the Minimum Matching Distance (MMD) for points sampled from our generated density volumes. Given a distance metric for two point clouds $X$ and $Y$, *e.g.* the Chamfer Distance (CD),

$$\text{CD}(X,Y) = \sum_{x \in X} \min_{y \in Y} \|x - y\|_2^2 + \sum_{y \in Y} \min_{x \in X} \|x - y\|_2^2, \tag{4}$$

COV measures the *diversity* of the generated point cloud set $S_g$, with respect to a reference point clout set $S_r$, by finding the closest neighbor in the reference set to each one in the sample set, and computing the fraction of the reference set covered by these samples:

$$\text{COV}(S_g, S_r) = \frac{|\{\arg\min_{Y \in S_r} CD(X,Y) | X \in S_g\}|}{|S_r|}. \tag{5}$$

MMD, in contrast, measures the the overall *quality* of these samples, by measuring the average distance between each sampled point cloud and its closest neighbor in the reference set:

$$\text{MMD}(S_g, S_r) = \frac{1}{|S_r|} \sum_{Y \in S_r} \min_{X \in S_g} CD(X,Y). \tag{6}$$

We compute these metrics for the PhotoShape Chairs and ABO Tables datasets, comparing our generated results to points sampled from the the same reference meshes used in the data splits from the evaluations in DiffRF [49]. For each generated object, we sample $2048$ points from a mesh extracted from the decoded density volume $V^{\text{Density}}$ (see Sec. 3.1) using the Marching Cubes [46] algorithm. We use a volume of resolution $64^3$ and $128^3$ for training the Chairs and Tables models, respectively. However, we note that downsampling these density volumes to $32^3$, as is used in DiffRF, before applying this point-sampling operation did not noticeably impact the results of these evaluations.

The results can be seen in Tab. 4, alongside the perceptual metrics from the main paper. Interestingly, these results show that, despite the increased flexibility of our approach, and DiffRF's restrictive use of both 2D rendering and 3D supervision on synthetic data when training their diffusion model, we obtain comparable or superior geometry compared to their approach, while substantially increasing the overall perceptual quality for these datasets. We also substantially outperform prior state-of-the-art approaches using GAN-based [6, 7] methods across both perceptual and geometric comparisons with these metrics.

Figs. 7 and 8 show qualitative comparisons between the unconditional generation results rendered using our method and DiffRF for each of these datasets. In each case, it is clear that for similar objects, our method produces more coherent and complete shapes without missing features, *e.g.* legs, and textures that are more realistic and detailed, leading to better and more consistent image synthesis results.

## A.2  Foreground Supervision

For some datasets with foregrounds with complex and varying appearance which can easily be mixed with the background environment, we found it necessary to supplement our primary autodecoder reconstruction loss (Sec. 3.2) with an additional foreground supervision loss. This loss measures how well depicted objects are separated from the background during rendering. To evaluate the effect of this foreground supervision, we ran experiments on the CelebV-Text [90] dataset both with and without this loss. We conduct our training until the autodecoder has seen a total of $9$ million frames from the training set, then reconstruct examples from the learned embeddings.

The result can be seen in Fig. 6. As depicted, the reconstructions without foreground supervision not only lack fidelity to the target appearance, but the estimated opacity and surfaces normals clearly show that the overall geometry is insufficiently recovered.

|        | PhotoShape Chairs [57] | | | | ABO Tables [13] | | | |
|--------|-------|-------|--------|--------|-------|-------|--------|--------|
| Method | FID ↓ | KID ↓ | COV ↑ | MMD ↓ | FID ↓ | KID ↓ | COV ↑ | MMD ↓ |
| $\pi$-GAN [6] | 52.71 | 13.64 | 39.92 | 7.387 | 41.67 | 13.81 | 44.23 | 10.92 |
| EG3D [7] | 16.54 | 8.412 | 47.55 | 5.619 | 31.18 | 11.67 | 48.15 | 9.327 |
| DiffRF [49] | 15.95 | 7.935 | 58.93 | **4.416** | 27.06 | 10.03 | **61.54** | 7.610 |
| Ours | **11.28** | **4.714** | **64.20** | 4.445 | **18.44** | **6.854** | 60.25 | **6.684** |

Table 4: **Quantitative comparison** of unconditional generation on the PhotoShape Chairs [57] and ABO Tables [13] datasets. Our method achieves a better perceptual quality, while maintaining similar geometric quality to the state-of-the-art diffusion-based approaches. MMD and KID scores are multiplied by $10^3$.

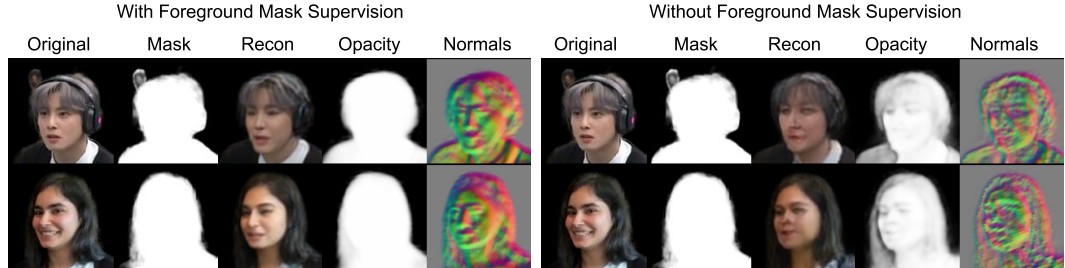

Figure 6: In real video datasets, *e.g.* CelebV-Text[90], we have a diverse set of foreground shapes and textures with a common background color. In these cases, we find that supervising the autodecoder with a foreground mask loss is important for the network to properly learn the shape of the object. Both examples shown after training for ∼9 million frames.

## A.3 Training the autodecoder on a single view per object

We conduct an experiment on training our autodecoder on the Chairs [57] dataset but only using a single view per object. To do so, we pick and train on a single rendering view among the 200 available, which is different for each object. Our autodecoder learns, nevertheless, to encode general geometry information and roughly infer the full 3D shape, of the object by utilizing information learned from other objects. However, it stills struggles with the texture details, and thus training with more training views is beneficial. We show the results on Fig. 9.

## A.4 Training a diffusion model on the embedding vector of the autodecoder

In the ablation section of our main paper, we experimented on different intermediate representation resolutions to train our diffusion model on. In Fig. 10 we show results of training only on the single vector we use as input to the autodecoder. Similarly to our $4 \times 4 \times 4$ run, we find the generates samples to be of low quality, failing to properly capture the geometry of the objects of the Chair dataset [57]. We hypothesize this compressed vector overfits to the training dataset, and diffusion struggles to fully capture the properties of the dataset distribution.

## A.5 Large and Diverge 3D Synthesis with Adversarial Methods

Adversarial 3D generation methods such as EG3D [7] and GET3D [21] have shown generative results of high perceptual quality when trained on single object categories. We experimented with trained them on Objaverse [14] our biggest and most diverse datasets, without utilizing any conditioning. In this setting, these methods fail to produce meaningful samples as we see in Fig. 11. In both cases, we see that the results stop to improve after a few epochs, with the FID never falling below 100, compared to our methods 40.49. We believe that without additional supervision such as class labels, these methods based methods will not produce good results.

DiffRF                                    Ours

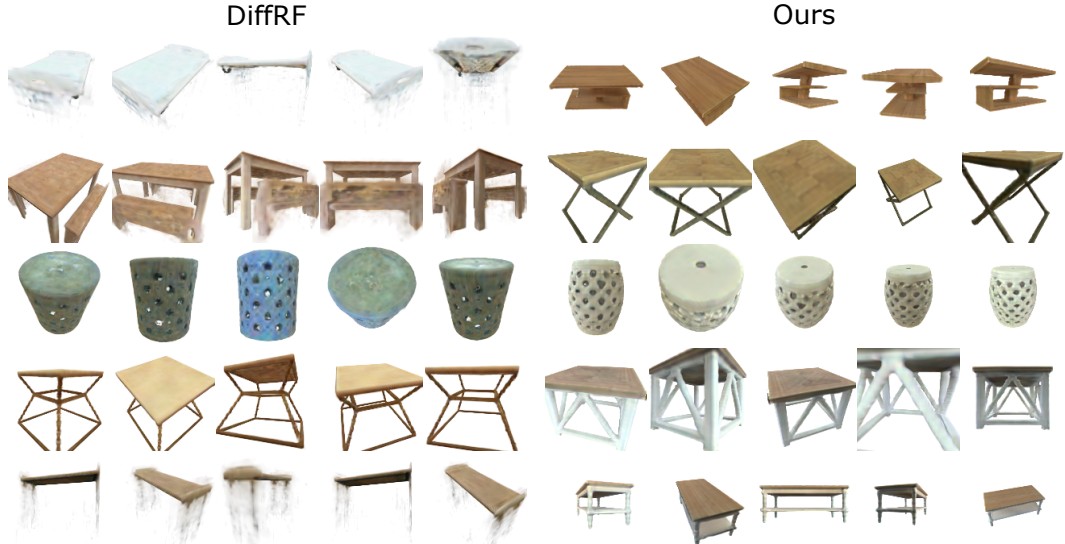

Figure 7: **Qualitative comparison of unconditional generation** using DiffRF [49] (left) and our approach (right) on the ABO Tables dataset [13]. In contrast to DiffRF, we train diffusion in the latent features of an autodecoder. Decoupling the expensive and demanding training from the output voxel-grid size lets us increase the resolution of our 3D representation. For this dataset, our output voxel resolution is $128^3$, compared to the $32^3$ resolution of DiffRF. Our method improves the perceptual quality of the results, as it as shown in the reported FID and KID.

DiffRF                                    Ours

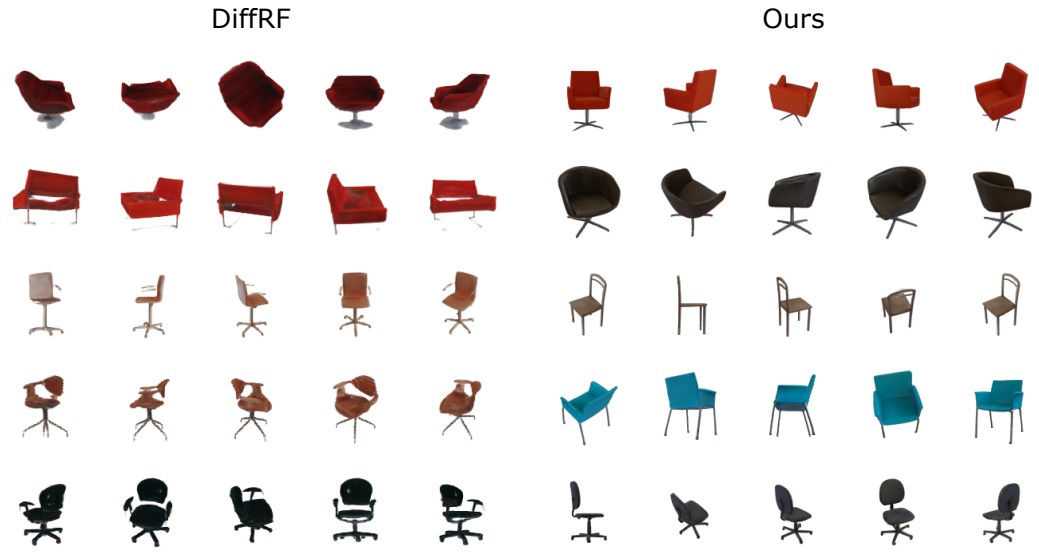

Figure 8: **Qualitative comparison of unconditional generation** using DiffRF [49] (left) and our approach (right) on the PhotoShapes Chairs dataset [57]. For this dataset, our output voxel resolution is $64^3$. As above, our results are both qualitatively and quantitatively superior.

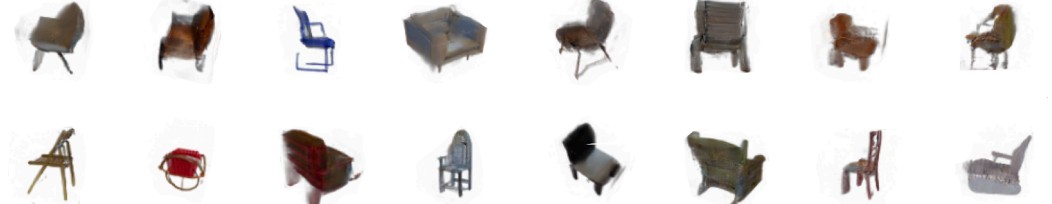

Figure 9: **Training the autodecoder on a single view per object on the Chairs dataset [57]**. the *autodecoder* offers a compressed representation of the dataset; it encapsulates prior knowledge. In contrast to a single-scene NeRF, our method can work with only a single view per object for single-category datasets. Our method can roughly learn the shape of the objects from multiple instances, but it struggles with textures and geometry details such as chair legs, so multiple views is still beneficial for precise reconstruction.

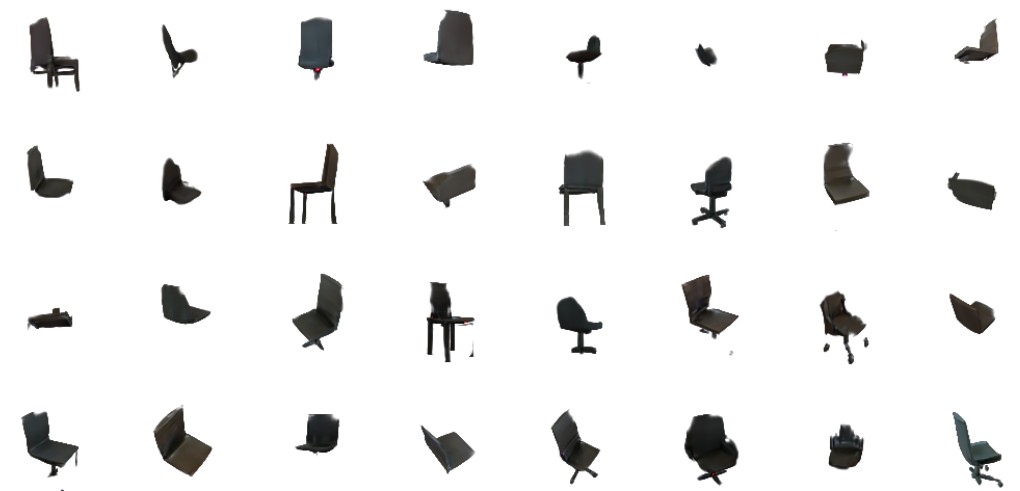

Figure 10: **Latent Diffusion Model trained on the embedding vectors of a pre-trained autodecoder**. Similarly, with our training on feature grid resolution of $4 \times 4 \times 4$, training on the vector embedding struggles to generate the concrete geometry of the objects in the Chairs dataset [57]

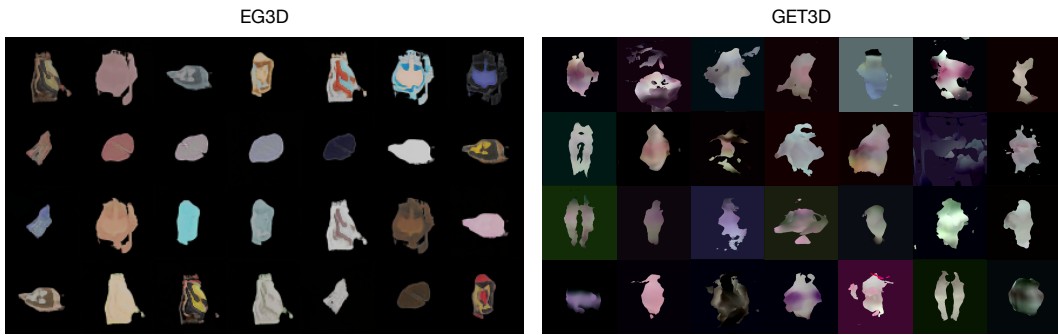

Figure 11: **Adversarial-based methods for large-scale and diverse 3D synthesis trained on Objaverse [14]**. While gan-based 3d generation methods EG3D [7] and GET3D [21] produce high-quality results for single object datasets, we observe that they strugle to converge in a large-scale and diverse setting such as Objaverse [14]

### A.6 Animated Results

Please visit our project's web page (`https://github.com/snap-research/3DVADER`) for additional video results, showing consistent novel-view synthesis for rigid objects from multi-category datasets and animated articulated objects sampled using our approach, and results demonstrating both conditional and unconditional generation.

## B  Method Details

### B.1  Volumetric Autodecoder

**Volumetric Rendering.** We use learnable volumetric rendering [48] to generate the final images from the final decoded volume. Given a camera intrinsic and extrinsic parameters for a target image, and the radiance field volumes generated by the decoder, for each pixel in the image, we cast a ray through the volume, sampling the color and density values to compute the color $C(\mathbf{r})$ by integrating the radiance along the ray $\mathbf{r}(t) = \mathbf{o} + t\mathbf{d}$, with near and far bounds $t_n$ and $t_f$:

$$C(\mathbf{r}) = \int_{t_n}^{t_f} T(t)\delta(\mathbf{r}(t))\mathbf{c}(\mathbf{r}(t), \mathbf{d})dt, \tag{7}$$

where $\delta$, $\mathbf{c}$ are the density and RGB values from the radiance field volumes sampled along these rays, and $T(t) = \exp\left\{-\int_{t_n}^{t} \sigma(\mathbf{r}(s))ds\right\}$ is the accumulated transmittance between $t_n$ and $t$.

To supervise the silhouette of objects, we also render the 2D occupancy map $O$ using the volumetric equation:

$$O(\mathbf{r}) = \int_{t_n}^{t_f} T(t)\delta(\mathbf{r}(t))dt. \tag{8}$$

We sample 128 points across these rays for radiance field rendering during training and inference.

**Articulated Animation.** As our approach is flexibly designed to support both rigid and articulated subjects, we employ different approaches to pose supervision to better handle each of these cases.

For articulated subjects, poses are estimated during training, using a set of learnable 3D keypoints $K^{3D}$ and their predicted 2D projections $K^{2D}$ in each image in an extended version of the Perspective-n-Point (PnP) algorithm [40]. To handle articulated animation, however, rather than learn a single pose per image using these points, we assume that the target subjects can be decomposed into $N_p$ regions, each containing $N_k$ points $K_p^{3D}$ points and their corresponding $K_p^{2D}$ projections per image. These points are shared across all subjects, and are aligned in the learned canonical space, allowing for realistic generation and motion transfer between these subjects. This allows for learning $N_p$ poses per-frame defining the pose of each region $p$ relative to its pose in the learned canonical pose.

To successfully reconstruct the training images for each subject thus requires learning the appropriate canonical locations for each region's 3D keypoints, to predict the 2D projections of these keypoints in each frame, and the pose best matching the 3D points and 2D projections for these regions. We can then use this information in our volumetric rendering framework to sample appropriately from the canonical space such that the subject's appearance and pose are consistent and appropriate throughout their video sequence. Using this approach, this information can be learned along with our autodecoder parameters for articulated objects using the reconstruction and foreground supervision losses used for our rigid object datasets.

As noted in Sec. 3.2, to better handle non-rigid shape deformations corresponding to this articulated motion, we employ volumetric linear blend skinning (LBS) [41]. This allows us to learn the weight each component $p$ in the canonical space contributes to a sampled point point in the deformed space based on the spatial correspondence between these two spaces:

$$x_d = \sum_{p=1}^{N_p} w_p^c(x_c)\left(R_p x_c + \mathrm{tr}_p\right), \tag{9}$$

where $T_p = [R_p, t_p] = [R^{-1}, -R^{-1}\,\mathrm{tr}]$ is the estimated pose of part $p$ relative to the camera (where $T = [R, \mathrm{tr}] \in \mathbb{R}^{3 \times 4}$ is the estimated camera pose with respect to our canonical volume) ; $x_d$ is the

3D point deformed to correspond to the current pose; $x_c$ is its corresponding point when aligned in the canonical volume; and $w_p^c(x_c)$ is the learned LBS weight for component $p$, sampled at position $x_c$ in the volume, used to define this correspondence. [3]

Thus, for our non-rigid subjects, in addition to the density and color volumes needed to integrate Eqns. 7 and 8 above, our autodecoder learns to produce a volume $V^{LBS} \in \mathbb{R}^{S^3 \times N_p}$ containing the LBS weights for each of the $N_p$ locally rigid regions constituting the subject.

We assign $N_k = 125$ 3D keypoints to each of the $N_p = 10$ regions. For these tests, we assume fixed camera intrinsics with a field-of-view of $0.175$ radians, as in [54]. We use the differentiable Perspective-n-Point (PnP) algorithm [40] implementation from PyTorch3D [62] to accelerate this training process.

As this approach suffices for objects with standard canonical shapes (*e.g.*, human faces) performing non-rigid motion in continuous video sequences, we employ this approach for our tests on the CelebV-Text dataset. While in theory, such an approach could be used for pose estimation for rigid objects (with only 1 component) in each view, for we find that this approach is less reliable for our rigid object datasets, which contain sparse, multi-view images from randomly sampled, non-continuous camera poses, depicting content with drastically varying shapes and appearances (*e.g.*, the multi-category object datasets described below). Thus, for these objects, we use as input either known ground-truth or estimated camera poses (using [66]), for synthetic renderings or real images, respectively. While some works [83, 42, 87] perform category-agnostic object or camera pose estimation without predefined keypoints from sparse images of arbitrary objects or scenes, employing such techniques for such data is beyond the scope of this work.

**Architecture.** Our volumetric autodecoder architecture follows that of [70], with the key extensions described in this work. Given an embedding vector $\mathbf{e}$ of size $1024$, we use a fully-connected layer followed by a reshape operation to transform it into a $4^3$ volume with $512$ features per cell. This is followed by a series of four 3D residual blocks, each of which upsamples the volume resolution in each dimension and halves the features per cell, to a final resolution of $64^3$ and 32 features. [4] These blocks consist of two $3 \times 3 \times 3$ convolution blocks each followed by batch normalization in the main path, while the residual path consists of four $1 \times 1 \times 1$ convolutions, with ReLU applied after these operations. After the first of these blocks we have the $8^3$ volume with 256 features per cell used for training our diffusion network, as in our final experiments. In this and the subsequent block, we apply self-attention layers [81] as described in Sec. 3.1. After the final upsampling block, we apply a final batch normalization followed by a $1 \times 1 \times 1$ convolution to produce the final $1 + 3$ density $V^{\text{Density}}$ and RGB color features $V^{\text{RGB}}$ used in our volumetric renderer.

**Non-Rigid Architecture.** For non-rigid subjects, our architecture produces $1 + 3 + 10$ output channels, with the latter group with the LBS weights for the $n_p = 10$ locally rigid components each region corresponds to in our canonical space. Our unsupervised 2D keypoint predictor uses the U-Net architecture of [69], which operates on a downsampled $64 \times 64$ input image to predict the locations of the keypoints corresponding to each of the 3D keypoints used to determine the pose of the camera relative to each region of the subject when it is aligned in the canonical volumetric space.

## B.2 Latent 3D Diffusion

**Diffusion Architecture and Sampling.** For our base diffusion model architecture, we use the Ablated Diffusion Model (ADM) of Dhariwal *et al.* (2021) [17], a U-Net architecture originally designed for 2D image synthesis. We incorporate the preconditioning enhancements to this model described in Karras *et al.* (2022) [36]. As this architecture was originally designed for 2D, we adapt all convolutions and normalizations operations, as well as the attention mechanisms, to 3D.

For the cross-attention mechanism used for our conditioning experiments, we likewise extend the latent-space cross-attention mechanism from Rombach *et al.* (2022) [63] to our 3D latent space.

**Robust Normalization.** Autoencoder-based latent diffusion models impose a prior to the learned latent vector [63]. We find the latent features learned by our 3D autodecoder already form a bell-like curve. However, we also observe extreme values that can severely affect the calculation of the

---

[3]In practice, as in [70], we compute an approximate solution using the inverse LBS weights following HumanNeRF [84] to avoid the excessive computation required by the direct solution.

[4]We add one block to upsample to $128^3$ for our aforementioned experiments with the ABO Tables dataset.

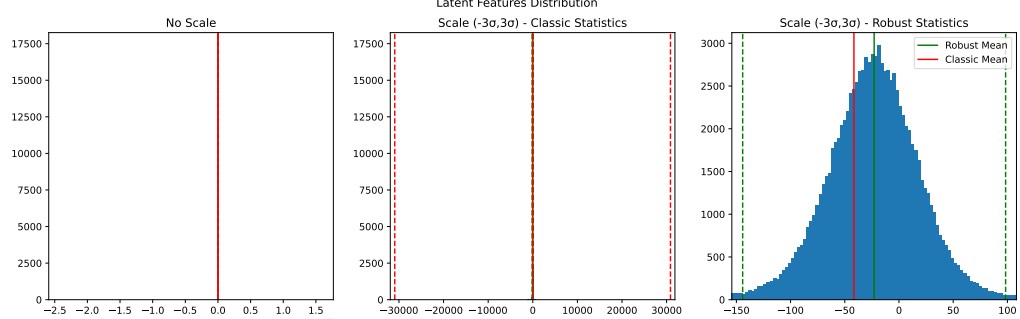

Figure 12: We present the latent feature distribution of a 3D *AutoDecoder* trained on MVImgNet[92]. The features are extracted at the $8^3$ resolution, where we apply diffusion. The three subplots show different levels of "zooming in." We see that the distribution spans a great range due to extreme outliers. Using classic mean and standard deviation computation, as we see in the middle subplot, still provides quite a large range of values. Normalizing the features using classic statistics leads to convergence failure for the diffusion model. We propose using robust statistics to normalize the distribution to $[-1, 1]$, before training the diffusion model. During inference, we de-normalize the diffusion output before feeding them to the upsampling layers of the autodecoder.

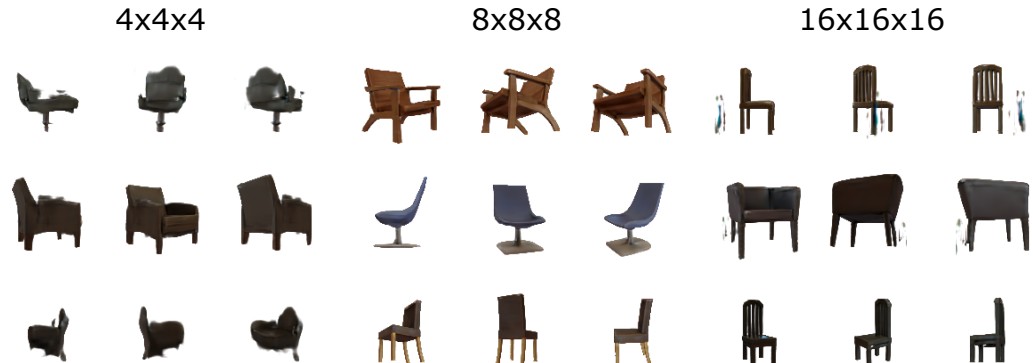

Figure 13: **Qualitative comparison of models trained at different latent resolutions.** All visualizations produced with 64 diffusion steps. We find that the model train on $8^3$ latent features gives the best trade-off between quality and training speed, rendering it the best option for training on large-scale 3D datasets.

mean and standard deviation. As discussed in the main manuscript, we deploy the use of *robust normalization* to adjust the latent features. In particular, we take the *median* $m$ as the center of the distribution and approximate its scale using the Normalized InterQuartile Range (IQR) [85] for a normal distribution: $0.7413 \times IQR$. We visualize its effect in Fig. 12. This is a crucial aspect of our approach, as in our experiments we find that without it, our diffusion training is unable to converge.

**Ablating the latent volume resolution used for diffusion.** We trained three diffusion models models for the same time, resources, and number of parameters, for diffusion at 3 resolutions in our autodecoder: $4^3$, $8^3$, and $16^3$. We find that the $4^3$ models, even when they train faster, often fail to converge to something meaning full and produce partial results. Most samples produced by the $16^3$ models are of reasonable quality. However, many samples also exhibit spurious density values. We hypothesize that this is due to the model being under-trained. The $8^3$ model produces the best results, and its fast training speed makes it suitable for large-scale training. We visualize the results in Fig. 13

### B.3 Hash Embedding

Each object in the training set is encoded by an embedding vector. However, as we employ multi-view datasets of various scales, up to ~300K unique targets from multiple categories, storing a separate embedding vector for each object depicted in the training images is burdensome [5]. As such,

---

[5]*E.g.*, the codebook *alone* would require *six* times the parameters of the largest model in our experiments.

we experimented with a technique enabling the effective use of a significantly reduced number of embeddings (no more than $\sim$32K are required for any of our evaluations), while allowing effective content generation from large-scale datasets.

Similar to the approach in [55], we instead employ concatenations of smaller embedding vectors to create more combinations of unique embedding vectors used during training. For an embedding vector length $l_v$, the input embedding vector $H_k \in \mathbb{R}^l$ used for an object to be decoded is a concatenation of smaller embedding vectors $h_i^j$, where each vector is selected from an ordered codebook with $n_c$ entries, with each entry containing collection of $n_h$ embedding vectors of length $l_v/n_c$:

$$H_k = \left[ h_1^{k_1}, h_2^{k_2}, ..., h_{n_c}^{k_{n_c}} \right],\tag{10}$$

where $k_i \in \{1, 2, ..., n_h\}$ is the set of indices used to select from the $n_h$ possible codebook entries for position $i$ in the final vector. This method allows for exponentially more combinations of embedding vectors to be provided during training than must be stored in learned embedding vector library.

However, while in [55], the index $j$ for the vector $h_i^j$ at position $i$ is randomly selected for each position to access its corresponding codebook entry, we instead use a deterministic mapping from each training object index to its corresponding concatenated embedding vector. This function is implemented using a hashing function employing the multiplication method [16] for fast indexing using efficient bitwise operations. For object index $k$, the corresponding embedding index is:

$$m(k) = [(a \cdot k) \bmod 2^w] \gg (w - r),\tag{11}$$

where the table has $2^r$ entries. $w$ and $a$ are heuristic hashing parameters used to reduce the number of collisions while maintaining an appropriate table size. We use 32 for $w$. $a$ must be an odd integer between $2^{w-1}$ and $2^w$ [16]. We give each smaller codebook its own $a$ value:

$$a_i = 2^{w-1} + 2 * i^2 + 1,\tag{12}$$

where $i$ is the index of the codebook.

**Discussion.** In our experiments, we found that employing this approach had negligible impact on the overall speed and quality of our training and synthesis process. During training the memory of the GPU is predominantly occupied by the gradients, which are not affected by this hashing scheme. For Objaverse, our largest dataset using $\sim$300K images, using this technique saves approximately 800MB of GPU memory.

Interestingly, this also suggests that scaling this approach to larger datasets, should they become available, will require special handling. Learning this per-object embedding would soon become intractable. However, simply using this *hash embedding* approach reduces the model storage requirements by $\sim$75% for this dataset.

In our experiments, we use hashing for ABO Tables, CelebV-Text and Objaverse, with codebook sizes $n_c =$ of 256, 8192 and 32768, respectively. We set the number of smaller codebooks ($n_h$) to 256 for each dataset.

## C   Implementation Details

### C.1   Dataset Filtering

**CelebV-Text [90].** Some heuristic filtering was necessary to obtain sufficient video quality and continuity for our purposes. We omit the first and last 10% of each video to remove fade-in/out effects, and any frames with less than 25% estimated foreground pixels. We also remove videos with less than 4 frames remaining after this, and any videos less than 200 kilobytes due to their relatively low quality. We also omit a small number of videos that were unavailable for download at the time of our experiments (the dataset is provided as a set of URLs for the video sources).

**MVImgNet [92].** For these annotated video frames depicting real objects in unconstrained settings and environments, we applied Grounded Segment Anything [39] for background removal. However,

as this process sometimes failed to produce acceptable segmentation results, we apply filtering to detect these case. We first remove objects for which Grounding DINO [45] fails to detect bounding boxes. We then fit our volumetric autodecoder (Secs. 3.1-2) to only the *masks* produced by this segmentation (as monochrome images with a white foreground and a black background). For objects that are properly segmented in each frame, this produces a reasonable approximation of the object's shape that is consistent in each of the input frames, while objects with incorrect or inconsistent segmentation will not be fit properly to the input images. Thus, objects for which the fitting loss is unsually high are removed.

**Objaverse [14].** While Objaverse contains ∼800K 3D models, we found that the overall quality of these varied greatly, making many of them unsuitable for multi-view rendering. We thus filtered models without texture, material maps, or other color and appearance properties suitable, as well as models with an insufficient polygon count for realistic rendering. Interestingly, given the simplicity of the objects when rendered against a monochrome background, we found that the foreground segmentation supervision used for the other experiments described in Sec. 3.2 of the main paper was unnecessary. Given the scale of this dataset (∼300K unique objects, with 6 frames per object), we thus omit this loss from our training process for this dataset for our final experiments for the sake of improved training efficiency. For datasets with more complex motion and real backgrounds, such as the real image datasets mentioned above, we found this supervision to be essential, as shown in Sec. A.2 and Fig. 6.

## C.2 Additional Details

**Training Details.** Our experiments are implemented in the PyTorch [58, 59], using the PyTorch Lightning [19] framework for fast automatic differentiation and scalable GPU-accelerated parallelization. For calculating the perceptual metrics (FID and KID), we used the Torch Fidelity [56] library.

We run our experiments on 8 NVIDIA A100 40GB GPUs per node. For some experiments, we use a single node, while for larger-scale experiments, we use up to 8 nodes in parallel.

We use the Adam optimizer [37] to train both the autodecoder and the diffusion Model. For the first network, we use a learning rate $lr = 5e - 4$ and beta parameters $\beta = (0.5, 0.999)$. For diffusion, we set the learning rate to $lr = 4.5e - 4$. We apply linear decay to the learning rate.

**Preparing the Text Embeddings for Text-Driven Generation.** We train our model for text-conditioned image generation on three datasets: CelebV-Text [90], MVImgNet [92] and Objaverse [14]. The two latter datasets provide the object category of each sample, but they do not provide text descriptions. Using MiniGPT4 [94], we extract a description by providing a *hint* and the first view of each object along with the question: "*<ImageHere></Img> Describe this <hint> in one sentence. Describe its shape and color. Be concise, use only a single sentence.*" For MVImgNet, this hint is the "class name", while it is the "asset name" for Objaverse.

Note this approach is not foolproof. To the contrary, we find that in many cases MiniGPT4 hallucinates descriptive characteristics of the object that do not match its visual input. We can see some examples like this in Fig. 14

With the text-image pairs for these three datasets, we use the 11-billion parameter T5 [61] model to extract a sequence of text-embedding vectors. The dimensionality of these vectors is 1024. During training, we fix the length of the embedding sequence to 32 elements. We trim longer sentences and pad smaller sentences with zeroes.

| | ABO-Tables | Chairs | CelebV-Text | MVImgNet | Objaverse |
|---|---|---|---|---|---|
| *3D AutoDecoder* | | | | | |
| $z$-length | 1024 | 1024 | 1024 | 1024 | 1024 |
| MaxChannels | 512 | 512 | 512 | 512 | 512 |
| Depth | 2 | 4 | 2 | 4 | 4 |
| SA-Resolutions | 8,16 | 8,16 | 8,16 | 8,16 | 8,16 |
| ForegroundLoss $\lambda$ | 10 | 10 | 10 | 10 | 0 |
| #Renders/batch | 4 | 4 | 4 | 4 | 4 |
| VoxelGridSize | $128^3 \times 4$ | $64^3 \times 4$ | $64^3 \times 14$ | $64^3 \times 4$ | $64^3 \times 4$ |
| Learning Rate | 5e-4 | 5e-4 | 5e-4 | 5e-4 | 5e-4 |
| *Latent 3D Diffusion Model* | | | | | |
| $z$-shape | $8^3 \times 256$ | $8^3 \times 256$ | $8^3 \times 256$ | $8^3 \times 256$ | $8^3 \times 256$ |
| Sampler | edm | edm | edm | edm | edm |
| Channels | 128 | 128 | 192 | 192 | 192 |
| Depth | 2 | 2 | 3 | 3 | 3 |
| Channel Multiplier | 3,4 | 3,4 | 3,4 | 3,4 | 3,4 |
| SA-resolutions | 8,4 | 8,4 | 8,4 | 8,4 | 8,4 |
| Learning Rate | 4.5e-5 | 4.5e-5 | 4.5e-5 | 4.5e-5 | 4.5e-5 |
| Conditioning | None | None | None/CA | None/CA | None/CA |
| CA-resolutions | - | - | 8,4 | 8,4 | 8,4 |
| Embedding Dimension | - | - | 1024 | 1024 | 1024 |
| Transformers Depth | - | - | 1 | 1 | 2 |

Table 5: Architecture details for our models for each dataset. *SA* and *CA* stand for *Self-Attention* and *Cross-Attention* respectively. $z$ refers to our 1D embedding vector and our latent 3D volume for the autodecoder and diffusion models, respectively. Note that for CelebV-Text, the output volume has 14 channels per cell: 3 for color values, 1 for density and 10 for part assignment.

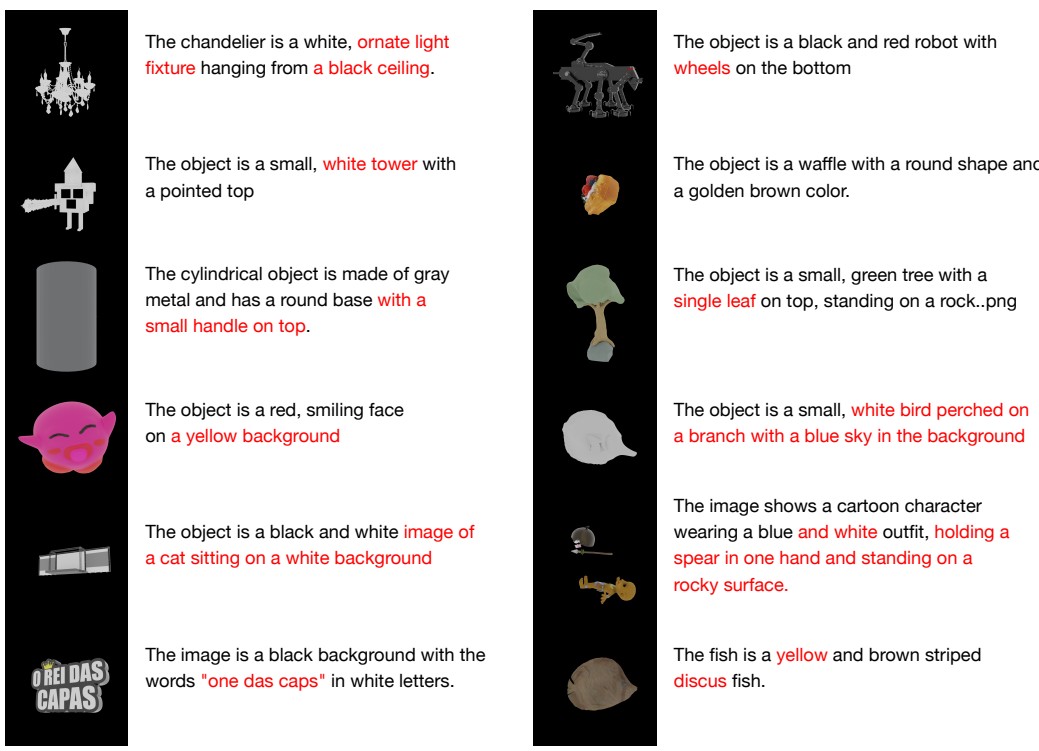

Figure 14: **Automated image captioning with MiniGPT4 [94] on MVImgNet [92]**. We can observe how the captioning tool hallucinated descriptions that do not match the input image, which leads to text-to-3D diffusion model learning not to adhere to every detail in the input prompt. We believe that better captioning systems should help alleviate this issue in the future

