# OpenReview forum: "Autodecoding Latent 3D Diffusion Models"
_NeurIPS.cc/2023/Conference — NeurIPS 2023 poster_

### Official Review · Reviewer_nmDk · 2023-06-22

**Soundness:** 3 good
**Presentation:** 3 good
**Contribution:** 3 good
**Rating:** 5
**Confidence:** 4

**Summary:**

In this manuscript, a novel method for unconditional and text-conditional generative models of 3D shape and texture representations is proposed. More specifically, the authors propose to train a 3D diffusion model of radiance and RGB fields that can be trained from 2D image and object mask supervision. The method comprises two stages, where in the first stage, an auto-decoder is trained, while in the second, a 3D UNet modeling a denoising diffusion process is trained on the latent features. At test time, random latent representations can be sampled and denoised to generate 3D radiance volumes that can be rendered from arbitrary viewpoints.

**Strengths:**

- The authors propose a valid method for training 3D diffusion models from posed 2D images. The key idea to first train an auto-decoder, and then in a second stage a 3D UNet describing a 3D denoising diffusion process is interesting and, with the success of recent latent 2D diffusion models in mind, an important field of study.
- The authors perform an extensive evaluation on 5 different datasets. To be able to run experiments on these different datasets, multiple automated masking and/or filtering steps needed to be performed (compare e.g. L. 252).
- The manuscript is well written, the technical explanations are sound, and the use of mathematical terms and symbols is correct.
- The manuscript contains helpful and well-organised figures such as the overview figure 1

**Weaknesses:**

- Experimental Evaluation in Table 1: Why does pi-GAN achieve such a high FID value of 52.71? Could the authors share some intuition on why the score is so bad? This seems not to be coherent with comparable prior works, for example GRAF [47] or GIRAFFE [43] report an FID of 34/20 on this dataset. Note that not improving over GAN-based approaches in FID on these single-object datasets is OK I believe, but it is important that the numbers are accurately reported.
- Qualitative Comparison: I believe the manuscript would benefit from a qualitative baseline comparison, e.g. for the methods reported in Table 1. It would further be interesting to discuss to a greater extent the differences of the results from the different generative model types, i.e. compare GAN-based results to the proposed diffusion model.
- The results are still quite limited, both qualitatively and quantitatively. I do appreciate that the authors tackle this hard task "feed-forward" of unconditional/text-conditional 3D generation, but e.g. the texture from the samples in Fig. 5 are very limited. I believe at least a text-based comparison and discussion to test-time optimization-based methods such as Dreamfusion is relevant here, as they lead to significantly better results. Potentially the proposed method could also be extended with an additional test-time-optimization step in future work.

**Questions:**

- Can the authors explain why the FID metric is so bad for the pi-GAN method?
- Can the authors explain what are the main reasons for the limited quality for the texture predictions in Figure 5?
- How important is it that the datasets contain multiple views (as opposed to only one) of the same object?

**Limitations:**

- The authors have discussed limitations as well as potential negative societal impact. As stated above, I believe the manuscript could benefit from a comparison/discussion against test-time-optimization-based methods due to the limited quality of the shown results.

---

> ### Author Rebuttal · Authors · 2023-08-10
>
> We would like to thank R.nmDk for their detailed response. We appreciate their highlighting of our work’s strengths. Namely, R.nmDk (a) finds our proposed 3D Diffusion model an interesting idea in an important field of study; (b) commends our extensive evaluation on different datasets, while appreciating the intricacy of their preparation; (c) appreciates our manuscripts writing, technical analysis and formulas, and well organized figures. Below, we try to address R.nmDk’s concerns and questions:
>
> **W1 and Q1 (pi-GAN FID):** We agree with R.nmDk on the importance of accurately reporting the numbers! For Table 1, we borrow all the metrics from DiffRF [a] (we will add this information on the table) , as we closely follow their dataset preparation and evaluation pipeline, which is different from GRAF’s. In particular, DiffRF renders “...15,576 chairs using Blender Cycles [13] from 200 views on an Archimedean spiral.” [a]. In contrast, GRAF renders “...150k Chairs from Photoshapes [49] following the rendering protocol of [46]” [b]. The dataset differences can vastly affect the output metric.
>
> **W2 (Qualitative comparison with baselines).** As R.nmDk points out, we do not include a visualization or discussion of diffusion-models and GAN-based models. We found that these qualitative results and analyses have been covered by DiffRF[a] and we focused on the differences of how we tackle the 3D Diffusion modeling problem compared to [a]. Similarly to DiffRF we observe that diffusion-based methods can achieve overall better quality and more meaningful structure and do not have view-dependent artifacts.
>
> For our qualitative comparison with DiffRF we would like to refer R.nmDk to Figures 7 and 8 in our supplement; for the discussion please see L51-55, L108-113, in the main text and L11-30 of the supplementary text.
> We appreciate R. nmDk’s suggestion and believe that including these comparisons will make our work more complete and comprehensive for the reader. We will add them in the final version of the manuscript.
> Summarizing these points, DiffRF is learning a single voxel grid for each object and is training a diffusion model on a dataset of learned voxel grids. This approach has several limitations: you need to save a Radiance Field for each item in the dataset: a time- and space-consuming process, especially for larger voxel grids. Additionally, 3D UNet training is very costly and thus prohibitive for larger voxel grids limiting the practical grid size.
>
> Our approach alleviates both of these problems. Our AutoDecoder represents the whole dataset as a collection of per-object embeddings plus the weights of a single decoder network. We run diffusion on a 8x8x8 grid which is significantly faster, and independent of the final output resolution. These changes permit our method to tackle large-scale datasets.
>
>
> **W3 (Comparison with Dreamfusion):** We discuss optimization-based methods in our related work section (L116-121). While we agree that Dreamfusion shows high-quality results, we disagree it is an apt comparison for our approach. They aim for generating a single object of high quality based on a text prompt; we aspire to enable large-scale 3D diffusion models. The main benefit is that Dreamfusion requires 1.5 hours on 4 TPU chips per object, while our method needs less than a few seconds to generate an object.
>
> In summary, we find Dreamfusion is tackling a different problem to ours. Nevertheless, we find that R.nmDk’s suggestion to combine the two approaches an excellent idea and would definitely explore it for future work. We believe score-distillation from a pretrained 2D model could not only help improve the texture quality, but also help distill additional knowledge to the one available in our 3D datasets.
>
> **Q2 (Texture prediction):** The Fig.5 contains objects generated from the model trained on images from the Objaverse dataset. We render these objects with uniform lighting, to exclude potential lighting inconsistencies. Because many objects in this dataset just have uniform texture, the model learned to use uniform color as texture for many objects (see sample from this dataset if Fig. C of the rebuttal pdf). Please, also refer to the supplementary material we show a more representative sample of the generated objects, many of which contain more complicated textures.
>
> **Q3 (Single view training):** As we mentioned above, the AutoDecoder offers a compressed representation of the dataset; it encapsulates prior knowledge. In contrast to NeRF, our method can work with Single-View for single-category datasets. Our method can roughly learn the shape of the objects from multiple instances, but it struggles with details such as chair legs, so multiple views is still beneficial for precise reconstruction. We show reconstruction results in Figure D of the rebuttal PDF. We are now running the diffusion stage and will add the results to the next version of the paper.
>
> [a] Diffrf: Rendering-guided 3d radiance field diffusion. CVPR 2023.
> [b] Graf: Generative radiance fields for 3d-aware image synthesis. NeurIPS 2020.
> [c] HoloDiffusion: Training a {3D} Diffusion Model using {2D} Images. CVPR 2023

---

> > ### Comment · Reviewer_nmDk · 2023-08-14
> >
> > I would like to thank the authors for the extensive and very informative rebuttal. The comparison w/ Dreamfusion was rather imagined as a discussion in the paper, i.e. pros and cons, and, as outlined, potential combinations; I agree that a side-by-side comparison of results is not required. I have no further questions at this point. Thanks!

---

### Official Review · Reviewer_iNSR · 2023-07-04

**Soundness:** 3 good
**Presentation:** 3 good
**Contribution:** 3 good
**Rating:** 7
**Confidence:** 4

**Summary:**

This paper presents a 3D generation framework that generalize to large-scale 3D dataset and articulated objects. The method comprises two parts. The first part is a 3D auto-decoder, reconstructing 3D objects from multi-view images or monocular videos. The second part is a latent diffusion model for unconditional or text-conditioned 3D generation. Extensive experiments are performed on 5 datasets, including the largest 3D dataset objaverse.

**Strengths:**

- I am impressed by the workload of this work. Not only are diverse static 3D objects are supported, but also articulated objects like 3D human heads.
- Experiments are solid. A total of 5 datasets are used, including Objaverse, which is the largest and the most challenging 3D dataset. Qualitative results on tables and chairs are reported and compared with 3D-aware GAN and 3D diffusion methods.
- Using the normalized IQR and median to normalize the latent features is an interesting trick. It is crucial for latent diffusion methods to deal with the latent normalization.

**Weaknesses:**

- The method itself is not novel, which I think is a minor issue since the main contribution of this work I think is the scaling up and generalizing the articulated objects.
- The quality of generated objects is inferior. I think this is limited by the reconstruction part. The resolution is limited to 64^3 due to the computational complexity of 3D volumes. I am wondering whether a CNN refinement/ super resolution module would help (e.g. EG3D, DiffRF)?
- The text-conditioned generation can only correctly generate the overall color and object category. Most detail descriptions are ignored. Other than the caption quality, is there any other possible reason for that, like the conditioning method design?

**Questions:**

N/A

**Limitations:**

Yes.

---

> ### Author Rebuttal · Authors · 2023-08-10
>
> We were delighted to read R.iNSR’s review! They appreciate the workload needed to achieve large and diverse static 3D object generation as well as synthesizing articulated human heads. Moreover, they commend our proposed Robust Normalization and De-Normalization scheme and its importance for latent diffusion modeling. In the following text, we aim to clarify the points they raised:
>
> **W2 (CNN refinement)**: Indeed CNN refiner could help to improve the visual fidelity of the results, although it may be at the expense of 3D consistency. We try autodecoding stage where we use CNN to refinement for ABO Tables dataset, we observe an improvement of the visual fidelity, however 3D geometry become more noisy. We provide visual results in the Fig. F of the rebuttal pdf. We will also add this experiment to the supplementary material.
>
> **W3 (Text Conditioning)**: We believe that the main problem is quality of the captions, mainly all the captions include some spurious details not related to the depicted content or do not describe detail at all, we provide some examples in the Fig. C of the rebuttal pdf, thus the model learns to completely ignore these details. Since we use exactly the same conditioning mechanism as Stable Diffusion, we think that this should not be a problem.

---

> > ### Comment · Reviewer_iNSR · 2023-08-16
> >
> > I thank authors for providing thorough rebuttals. My concerns are addressed. Therefore, I would keep my rating.

---

### Official Review · Reviewer_xDwW · 2023-07-06

**Soundness:** 4 excellent
**Presentation:** 3 good
**Contribution:** 3 good
**Rating:** 6
**Confidence:** 4

**Summary:**

This paper proposes a diffusion model that learns to generate 3D objects, using only multi-view images or videos for training. It first trains a 3D convolutional autodecoder to embed the dataset; this maps latent vectors via a 3D feature space to voxelised scenes, and is trained for reconstruction of volumetrically-rendered images. Then, it trains a diffusion model over the feature space of this autodecoder, to enable a-priori generation; there are experiments on which layer of 3D features are best to use. The method is demonstrated on a variety of datasets, both synthetic and real, including rigid objects such as chairs, and non-rigid objects such as human faces.

**Strengths:**

The proposed pipeline is novel, its components are clearly described, and most design decisions are well motivated/justified (and appear sound). There are insightful discussions on how the autodecoder architecture affects performance.

The evaluation is rather comprehensive, covering five datasets of somewhat different character. Some of these are synthetic and use ground-truth camera poses, while others use imperfect poses from SfM.

Generation results on synthetic images (chairs & tables) are quantitatively better than baselines – both FID and KID are lower than pi-GAN, EG3D, and DiffRF. Qualitative results here also look good.

Generation results on real images are found to be quantitatively of higher quality (lower FID/KID) than those generated by naively sampling in the latent space of the autodecoder (without any diffusion process).

There is an ablation study covering various important design decisions in the autodecoder (mainly architectural choices), and an additional set of experiments investigating which feature layer of the autodecoder the diffusion is performed over, and how many diffusion steps are used.


**Weaknesses:**

While it may be accurate that "no prior work demonstrates the ability to generalize to
such large-scale datasets" (L282), I feel there should be some attempt at a quantitative comparison here – e.g. retraining the best of the prior works on these datasets to see how well or badly they perform. In particular, Objverse, CelebV-Text and MVImgNet are all extremely recent, so it may simply be that the prior works have not been tested on those datasets (as they were not yet available), but would still work to some degree. This is particularly important to put the (rather high!) FID scores of tab. 3 in context. An alternative that would go some way to mitigating this problem would be to use one of the more-constrained by still photographic datasets used in pi-GAN or EG3D (e.g. human/cat faces or cars), and evaluate how well the proposed method performs on this.

Qualitative results on MVImgNet (only given in the supplementary) are not very impressive – it is often impossible for this reviewer to determine the class of the generated objects. Similarly, the results on face generation (with a target) look significantly lower fidelity than other recent methods like EG3D.

The qualitative text-conditioned generation results in fig. 4 & 5 are not particularly impressive. Moreover, the 'ground-truth' text labels are from a pretrained captioning model, and thus noisy. It would be valuable to include an experiment with high-quality captions, so the impact of caption quality vs model power can be understood.

The technical contribution is a little small for NeurIPS – both latent diffusion and autodecoding of 3D shapes are pre-existing techniques, and their composition does not seem to be particularly complicated. This is somewhat mitigated by the detailed experiments and extensive discussion of design choices.


**Questions:**

Why is an autodecoder approach chosen, instead of (for example) VQ-GAN or autoencoding?

See above under weaknesses regarding performance comparison with baselines on real-world data – this is my largest concern.

How are the 'driven' face animations in fig. 2 generated? I didn't see this described anywhere in the text.


**Limitations:**

There is adequate discussion of both limitations and broader impact.

---

> ### Author Rebuttal · Authors · 2023-08-10
>
> We would like to thank R.xDwW for the informative review. We are delighted that they found our pipeline novel and our evaluation comprehensive. R.xDwW also highlights the comparison with respect to the baselines and ablation study. In the following we address R.2ZWE points:
>
> **W1 and Q2 (Prior methods on large scale dataset):** We agree with the R.xDwW, that comparison of prior gan-based methods on large scale dataset such as objaverse will be useful. To this end we launch a training run of EG3D on Objaverse dataset. Since the dataset is very large, training is only at about 50%, the best FID was 105.15 (that was reached at 20% of the training) while FID for our method is 40.49, the visual results are provided in the Fig. A of the rebuttal pdf. We will provide final results in the next version of the paper, however we observe that FID stops improving at some point and now oscillates at around 120, thus we believe it will not become much better after this point.  We advocate this behavior to the notorious difficulty of training GAN-based with diverse categories, without additional supervision such as class labels. We believe this experiment already demonstrates the scalability issues of the GAN based models. We would like to highlight that running EG3D on MvImgNet will be even more problematic, since the camera distribution in this dataset is unknown and depend on the considered object. For example some clothing items may be only shown from frontal view, while some toys may only shown from the top. CelebV, on the other hand, has articulated objects which EG3D does not support.
>
> **W2 (Qualitative results)**: We agree with the R.xDwW, that it may sometimes be hard to understand the class of the object from generated results in MvImgNet, however we believe that this is partially a dataset issue. This dataset features a lot of fruits and vegetables and their pieces which can be hard to recognize. We provide a sample from this dataset in Fig. B of the supplement rebuttal. Regarding CelebV-Text, while indeed the visual fidelity is lower than EG3D we would like to highlight that the considered setting is very different, we target articulated objects while EG3D targets static ones. Moreover, unlike EG3D we did not use ground truth poses, so a fair comparison would be EG3D w/o cameras where the face geometry is completely flat (see Fig.4 in EG3D supplementary material).
>
> **W3 (High quality captions)**: We fully agree that the method will hugely benefit from better captions, however at the time of the submission no caption for the MvImgNet and Objaverse was available, thus we resort to off-the-shelf captioning system. After the submission, one of the concurrent works [a] proposed a method for annotating the Objaverse dataset; we plan to utilize these captions for a future work.
>
> **Q1 (Why autodecoder)**: The problem of both VQGan and autoencoder, is the encoder part. The encoder assumes that the output of the autoencoder is already known in advance, which does not work for 3D generation, since for most of the real world datasets we only have images. There are several methods [b, c] that work with images as input, but the reconstruction quality of these works is significantly lower compared to optimization approaches. Another disadvantage of these approaches is that they can only be trained with a small number of views, thus it hard to utilize datasets with large number of views such as PhotoShape Chairs and ABO Tables (about 100 views) and CelebV-Text (up to 1000 frames for each video).
>
> **Q3 (Driven animation)**: For driving animation we generate an animatable asset form our diffusion model and then we animate it with poses extracted by our pose predictor from driving video shown at the left. We will explain this in more details in supplementary material, for more technical details on how articulated generation is performed please refer to **Sec. B1** of the supplementary material.
>
> [a] Scalable 3D Captioning with Pretrained Models - Arxiv 2023
> [b] pixelNeRF: Neural Radiance Fields from One or Few Images - CVPR 2021
> [c] ViewFormer: NeRF-free Neural Rendering from Few Images Using Transformers - ECCV 2022

---

> > ### Comment · Reviewer_xDwW · 2023-08-15
> > **Post-rebuttal**
> >
> > Thanks for the detailed rebuttal. This resolves some of my concerns, and overall I remain positive about this paper.

---

### Official Review · Reviewer_2ZWE · 2023-07-06

**Soundness:** 3 good
**Presentation:** 2 fair
**Contribution:** 2 fair
**Rating:** 5
**Confidence:** 3

**Summary:**

The paper proposes a 3D autoencoder to learn a latent volumetric space on the training dataset, which can be decoded into a radiance volumetric representation for novel-view synthesis, and then learn a 3D diffusion model on the latent volumetric representation. The latent volumetric space is acquired by training NeRF on multiple views (or frames).

**Strengths:**

- Diverse datasets, including synthetic and real-world, rigid and articulated objects, are used for evaluation.

**Weaknesses:**

1. Some claims in the paper are not verified by the evidence. See *Questions* for examples.
2. Only novel-view synthesis is demonstrated. Since the authors use a volumetric representation, the method is restricted by the voxel resolution. The visualized images look small and coarse. It will be better if the authors can show some 3D results (e.g., extracting meshes from radiance fields). Thus, compared to prior and concurrent works (see *Questions*), this method does not seem to be more scalable or extendable. Especially, the voxel resolution is a bottleneck.
3. Camera poses still seem to be necessary in this work, either automatically being annotated in synthetic data or estimated for in-the-wild data. It is a little tricky to say that this work does not need "3D supervision". The authors can do such an experiment to convince the readers that the proposed method is robust to camera poses: comparing a model trained with estimated poses on synthetic data and one trained with GT poses.


Minor typo:
- L212: One of our key observation -> One of our key observations?

**Questions:**

1. In L41-L43, the authors claim that "our approach is thus designed to be robust to" different sources of poses. However, I do not find relevant explanations in the method or experiment section. Can the authors elaborate on it?
2. In L150-L151, the authors claim that "intermediate representations such as feature volumes or tri-planes, as it is more efficient to render and ensures consistency across multiple views". However, several prior or concurrent works (e.g., 3D Neural Field Generation using Triplane Diffusion, 3DGen: Triplane Latent Diffusion for Textured Mesh Generation, Single-Stage Diffusion NeRF: A Unified Approach to 3D Generation and Reconstruction) show that feature volumes or tri-planes can work well. Can the authors provide any ablation study to support the claim?
3. In L62, "To identify the best scale" is unclear, as the word "scale" is not mentioned before.
4. It is unclear which datasets are used for training and evaluation. In L71-76, it is unclear whether the authors train 3 models on 3 datasets separately or they train a model on 3 datasets progressively. And in Sec 4.1, it is unclear whether all datasets are just for evaluation, or some of them are used for training. The authors should clarify their training and evaluation protocols.

**Limitations:**

The authors have adequately addressed the limitations.

---

> ### Author Rebuttal · Authors · 2023-08-10
>
> We would like to thank R.2ZWE for their thought-provoking review. We are glad they appreciate that our method works on diverse datasets, both real-world and synthetic, as well as rigid and articulated objects. Let us address R.2ZWE points:
>
> **Q4 (Training and Evaluation Protocol):** Regarding evaluation and training. We follow the evaluation protocol of DiffRF, we train on one dataset and then compute FID and KID on the same dataset. In more detail we train 5 different models, for each of the five datasets considered in this paper: PhotoShape Chairs, ABO Tables, CelebV-Text, MVImgNet and Objaverse. Everything in these experiments is separated and they don’t have any interactions of any kind.
>
> **W3 and Q1 (Camera poses):** Regarding the camera poses, we use different sources of the camera poses for different datasets. Predicting camera poses is an extremely challenging task, and we are not claiming to provide a solution for it in this work. This is especially challenging for arbitrary object categories. We claim that our method could work with different sources of camera pose:
> With ground truth camera poses that are available in synthetic datasets: PhotoShape Chairs, ABO Tables and Objaverse.
> With datasets of rigid objects where COLMAP can provide reliable estimates of the camera, such as MvIMGNet.
> Non-rigid objects where we train a camera prediction model (which in this case also acts as pose prediction model) together with the autoencoder, without any additional supervision or COLMAP predictions, see CelebV.
> So hence we are saying that "our approach is thus designed to be robust to" different sources of poses, in the meaning “our approach is flexible to work with different sources of poses”. Here we did not claim that it is “robust to pose estimation errors from COLMAP”. We are sorry for the confusion and we try to make it clearer in the final version.
>
> **W2 and Q2: (Concurrent Triplane works)**  We would like to bring to R.2ZWE attention the fact that all the works R.2ZWE mentioned [a, b, c] were not published during the time of the submission, and thus are concurrent works. Because of this we did not include comparison with them in our original manuscript. We provide extracted meshes in the Fig. H (rebuttal pdf).
> As R.2ZWE correctly mentioned, all these works use triplanes as intermediate representation, and we agree that theoretically Triplanes can be used for datasets where a lot of views for supervision is available, such as objaverse. When the multi-view supervision is scarce and ground truth camera information is not available, such as in video datasets like CelebV, Triplanes tend to degrade to prediction of the flat objects, which was observed in the prior work [d]. Another disadvantage of Triplanes and feature volumes is additional MLP requirement, this MLP significantly increases time for each individual forward pass. We provide a timing comparison below:
>
> | Representation | Time for 1 iteration (s) |
> | :----: | :----:|
> | Voxel Grid $4\times64^3$     | 0.22|
> | Triplane     $96\times64^2$   | 0.33|
> | Triplane     $96\times128^2$ | 0.33|
> | Triplane     $96\times256^2$ | 0.38|
> | Triplane     $96\times512^2$ | 0.46|
>
>
> Here we render at 128x128 resolution with 128 points per ray, for Triplane we use 2 layer MLP with 32 hidden neurons and the triplane generator uses exactly the same architecture as our Voxel generator. While the generation of the Triplane is relatively lightweight, the MLP is very heavy. Coming back to the concurrent works mentioned by R.2ZWE, only [b] was shown to be trained on a large scale multi category dataset. This is achieved by training a latent diffusion model with auto**encoder** where the input to it is dense point cloud, autoencoder need sufficiently less iteration to converge thus it is feasible to train with Triplanes as intermediate representation. However the requirement of dense point clouds require ground truth object meshes, which is not available for the datasets like MVImgNet or CelebV. Thus our method covers significantly more possible scenarios than [b].
>
> **Q3 (best scale):** Thank you for pointing this out, here it should be “To identify best  intermediate representation …”.
>
> **W3 (observations typo):** Thank you, we will fix the typo.
>
> [a] 3D Neural Field Generation using Triplane Diffusion. CVPR - 2023.
>
> [b] 3DGen: Triplane Latent Diffusion for Textured Mesh Generation. Arxiv - 2023. (Probably ICCV - 2023)
>
> [c] Single-Stage Diffusion NeRF: A Unified Approach to 3D Generation and Reconstruction.  ICCV - 2023
>
> [d] Unsupervised Volumetric Animation, CVPR - 2023

---

> > ### Comment · Reviewer_2ZWE · 2023-08-14
> >
> > Thank the authors for their reply. My concerns about training and Evaluation Protocol as well as camera poses have been resolved, which enhances the strength of this paper as it can handle different datasets with different sources of poses and multi-view images.
> >
> > - For related works using tri-plane representation, I am not sure whether it is reasonable to claim that [a] (CVPR 2023) is a concurrent work. But it will not affect my rating.
> > - "When the multi-view supervision is scarce and ground truth camera information is not available, such as in video datasets like CelebV, Triplanes tend to degrade to prediction of the flat objects, which was observed in the prior work [d]." I agree with this claim. However, I wonder whether and why the voxel representation trained using rendering representation can handle this.
> > - I actually appreciate the authors' efforts to add visualizations of extracted meshes. However, can the authors also extract and show colors from NeRF representation as well?

---

> > > ### Author Response · Authors · 2023-08-21
> > > **Reply to Reviewer 2ZWE**
> > >
> > > We are happy that R.2ZWE concerns about training and evaluation protocol as well as camera poses have been resolved. We will try to address the rest of the questions:
> > >
> > > ***Regarding Triplane’s flat geometry solution:***
> > >
> > > This is an observation made in [d]. Our intuition is that tri-planes have a design bias towards flat surfaces due to their plane representation. Voxels, however, do not have this, due to their true 3D structure, and thus are more robust against flat-surface local minima.
> > >
> > > ***Regarding the colored meshes:***
> > >
> > > We thank R.2ZWE for the suggestion, we will add more visualizations in the supplementary material.
> > >
> > > Please find some unconditional diffusion results in this anonymous image link:
> > > https://imgur.com/a/wwMORpv
> > >
> > > [d] Unsupervised Volumetric Animation, CVPR - 2023

---

### Official Review · Reviewer_Nn3Q · 2023-07-16

**Soundness:** 2 fair
**Presentation:** 2 fair
**Contribution:** 2 fair
**Rating:** 6
**Confidence:** 4

**Summary:**

In this paper, the authors propose an approach to learning a latent diffusion model for 3D assets from image (or video) supervision. The core is that they use an autodecoder architecture, which learns embeddings in the latent space by decoding them into a volumetric representation for rendering. Then, the authors identify the appropriate intermediate volumetric latent space and introduce a normalization technique to learn a latent diffusion model. In addition, this method can use either existing camera information or no camera information during training. The experiments demonstrate the proposed method is able to generate both static and articulated 3D objects.

**Strengths:**

- The proposed method has a broader application domain and is extendable to large-scale datasets. It can be applied to various datasets, including rigid static and articulated moving objects.
- The proposed modifications are effective in improving the 3D autodecoder.
- The authors propose a robust normalization trick to train latent diffusion models on every intermediate volumetric latent space.

**Weaknesses:**

- The authors miss a proper discussion with an important related work, GAUDI [a]. GAUDI first learns a 3D autodecoder from images and then trains a diffusion model on the latent space as a prior for generation. There are several similar points between this work and GAUDI. However, GAUDI is not even cited.
- The learned 3D voxel grids are treated as "Canonical Representation." However, how to properly define a canonical view of large-scale 3D datasets? For example, in Objaverse, the object poses are not well-aligned. Even for a single category (e.g., chair), the objects are randomly placed. Will this case influence the performance of the proposed method? Do you have any observations?
- Considering the autoencoder-based method needs to regularize the bottleneck, the autodecoder can also regularize their bottleneck as done in [c, d]. Have you tried to train a 1D diffusion model (see [e]) in the latent space of the autodecoder to see the generation ability?
- Missing comparison with some important baselines, such as GET3D [f]. I also suggest the authors train GET3D and DiffRF for Table 3 for a fair comparison.
- I have some questions or suggestions according to the writing:
  - In Line 1, the authors assume diffusion models are all latent diffusion models. I suggest the authors polish their writing to avoid this misuse.
  - In Line 57, the authors mention, "First, our autodecoders do not have a clear "bottleneck." Is the bottleneck here a latent space? If it is the case, the embedding space is the latent space of the autodecoder.
  - In Line 147, it is suggested to add some credits to prior work on trilinear interpolation for volumetric rendering, such as Plenoxels [b].
  - Can you elaborate more on the direct latent sample?
  - It is suggested to move the "Hash Embedding" part to the main paper. Otherwise, the vanilla autodecoder approach can not be scaled to large-scale datasets due to the large size of the embedding.
  - It is suggested to add [a-e] to the reference.
- In LDM [46], they use a normalization technique on the latent space. The proposed robust normalization seems to be an extended version of this.

[a] GAUDI: A Neural Architect for Immersive 3D Scene Generation. NeurIPS 2022.

[b] Plenoxels: Radiance Fields without Neural Networks.CVPR 2022.

[c] Demystifying Inter-Class Disentanglement. ICLR 2020.

[d] Rethinking Content and Style: Exploring Bias for Unsupervised Disentanglement. ICCV 2021.

[e] Latent Point Diffusion Models for 3D Shape Generation. NeurIPS 2022.

[f] GET3D: A Generative Model of High Quality 3D Textured Shapes Learned from Images. NeurIPS 2022.

**Questions:**

Please refer to the weakness section. I will adjust my rating according to the rebuttal.

**Limitations:**

The limitation and broader impact part look adequate to me.

---

> ### Author Rebuttal · Authors · 2023-08-10
>
> We would like to thank R.Nn3Q for their review. We appreciate their highlight of broader method application, effectiveness of our 3D autodecoder modifications and application of robust normalization. Next we answer weakness and questions:
>
> **W1 (GAUDI Discussion):** Thank you for pointing that out, GAUDI[a] is indeed a relevant work, we will add citation and discussion into the final version of the paper. We would like to highlight a couple of differences between our work and [a]. GAUDI is used for a pretty narrow application - indoor scene fly-through generation. The datasets in this application are relatively small, from 18 to maximum 1.8k scenes. While in our work we try to tackle datasets of 2 orders of magnitude higher. This work also does not model articulated objects. Moreover the diffusion process is trained in the z space of the auto-decoder, which we later show has significantly lower performance compared to the space with spatial arrangement, e.g. $8^3$ voxel, in our setting.
>
> **W2 (Canonical Object Orientation):** The canonical object orientation is indeed important for autodecoder fitting. The better alignment objects have, the easier it is to fit an auto-decoder. To demonstrate that we run the experiment of PhotoChairs dataset, where we randomly flip some of the objects upside down. The fitting error in terms of perceptual loss is 0.76 versus 0.52 (for original dataset). Thus significant effort was made towards pre-processing large scale datasets. For an objaverse dataset we observe that most of the objects are axis aligned, thus we just center at (0,0) and rescale according to the largest axis. In MVImgNet, there is no canonical orientation, thus we instead use partial sparse point clouds provided with a dataset to center and then perform PCA to orient. On the other hand, canonicalization for CelebV is obtained automatically, because of the joint camera estimation and object reconstruction. We believe that automatic canonicalization for a more general dataset, such as MVImgNet could be interesting future work.
>
> **W3 (1D Diffusion model):**. We find that because autodecoder is trying to compress large scale dataset into relatively small latent space, the early layers act more like a dictionary, not as an upsampler, because of this it maybe hard for the diffusion to operate in the early layers. We show this phenomenon for $4^3$ representation in Fig.3 of the main paper. Additionally we run an experiment with 1D representation; the visual samples are provided in Fig. G of the rebuttal pdf. We observe that at this stage the model behavior is very similar to direct latent sampling as the method fails to produce correct object geometry.
>
> **W4 (Get3D comparison):** Since we already compare with two GAN-based baselines, we did not include Get3D. However we scheduled a training run on Objaverse, the preliminary results are 259.18 FID (see Fig. F of supplementary pdf for visual results), which is significantly larger than our model with 40.49. We believe this is because GAN-based methods struggle with fitting diverse categories, without additional supervision such as class labels, Get3D will not produce good results. We still plan to finish the training on this dataset and add this experiment in the supplementary material, but we do not expect it to perform significantly better than this. We also would like to point out that DiffRF requires fitting a NeRF for 300k objects as a preprocessing stage, which is infeasible with our current resources. Running DiffRF on MvImgNet will not work, since in this dataset most of the objects have only partial visibility. For example some clothing items may be only shown from frontal view, while some toys may only shown from the top. This issue will also complicate application of Get3D, since it is not trivial to devise appropriate camera distribution. CelebV, on the other hand, has articulated objects which both Get3D and DiffRF do not support.
>
> **W5 (All Diffusion models are latent):** We fixed this in the new version of the paper.
>
> **W5 (Autodecoder bottleneck):** Sorry for the confusion, we used the “bottleneck” term for “latent representation on which diffusion operates”, since for other latent diffusion models they are the same, but we agree that in our case it makes sense to use a different term. We will change this in the future version of the paper.
>
> **W5 (Plenoxels):** Thank you for the suggestion, we will add the proposed work.
>
> **W5 (Direct Lattent Sampling):** The direct latent sampling is basic multivariate gaussian fitting on the 1D embeddings for the entire dataset. This method was used in prior work [1]. In case of hash embedding we follow [a] and randomly sample the indices of each hashtable.
>
> **W5 (Hash Embedding):** According to your suggestion, we will shortly describe hash embedding in the main paper, and keep the details for supplement.
>
> **W5 (Citations):** We will add all the proposed methods to the related work.
>
> **W6 (LDM normalization vs robust normalization):** Similarly to LDM we confirm that normalization of the latent space is a crucial part of the diffusion training. However we want to clarify that we did not claim this as our finding. But we disagree that proposed robust normalization is an extension of LDM normalization. LDM normalization is based on KL minimization during training, this is used as an additional loss, which may hurt the reconstruction. Instead our robust normalization can be applied after the training, thus does not affect the reconstruction in any way. Another benefit of the proposed normalization is the ability to select the representation on which to operate after the autodecoder training.
>
> [a] StyleGenes: Discrete and Efficient Latent Distributions for GANs. ArXiV 2023

---

> > ### Comment · Reviewer_Nn3Q · 2023-08-21
> >
> > Thanks for your detailed rebuttal. I decide to raise my rate to WA. Please remember to add the promised content if accepted.

---

### Author Rebuttal · Authors · 2023-08-10

We would like to thank all the reviewers for their thoughtful and detailed reviews. We are delighted to see that they found our method broadly applicable (R.Nn3Q), our modification effective (R.Nn3Q) and evaluation extensive (R.2ZWE, R.xDwW, R.iNSR, R.nmDk). It is also nice to see that reviewers appreciate the importance of our proposed robust normalization technique (R.Nn3Q, R.iNSR). We also would like to thank R.nmDk for appreciating the quality of writing and figures, and both R.nmDk and R.iNSR for highlighting the workload needed to handle experiments on such a diverse set of large scale datasets. We provide the response to each individual reviewer in the corresponding section.

---

> ### Author Response · Authors · 2023-08-21
> **Post-Discussion**
>
> We would like to thank all the reviewers for taking the time to reply to our responses. We are glad that: they found our rebuttal detailed (R.Nn3Q), extensive and informative (R.nmDk); reassured or improved their positive outlook for the paper (R.xDwK, iNSR, Nn3Q); and had their concerns addressed (R.2ZWE, R.xDwW). We will update our manuscript, adding all the additional content promised in our responses.

---

### Decision · Program_Chairs · 2023-09-21

**Decision:**

Accept (poster)

**Comment:**

The paper proposed a audodecoding framework with a diffusion model on voxel latent space for the 3D generative model, and showed broad applicabilities of the method, including generating rigid and articulated objects, using real and synthetic datasets, with synthetic or estimated cameras. The paper also proposed some effective components, including robust normalization, and diffusion models design on the voxel latent. The rebuttal and discussions addressed the concerns of the additional comparisons with other baselines. Considering the weaknesses and strengths, all the reviewers are becoming positive about this paper. I follow the reviewers' consensus and would recommend accepting this paper.

Please add the additional comparisons and discussions to other baselines to the main paper, the motivation of why voxel-based representation is better than tri-plane representation, as well as the writing clarifications mentioned by the reviewers.